# ChA-MAEViT: Unifying Channel-Aware Masked Autoencoders and Multi-Channel Vision Transformers for Improved Cross-Channel Learning

**Chau Pham**
Boston University
chaupham@bu.edu

**Juan C. Caicedo**
Morgridge Institute, UW–Madison
juan.caicedo@wisc.edu

**Bryan A. Plummer**
Boston University
bplum@bu.edu

## Abstract

Prior work using Masked Autoencoders (MAEs) typically relies on random patch masking based on the assumption that images have significant redundancies across different channels, allowing for the reconstruction of masked content using cross-channel correlations. However, this assumption does not hold in Multi-Channel Imaging (MCI), where channels may provide complementary information with minimal feature overlap. Thus, these MAEs primarily learn local structures within individual channels from patch reconstruction, failing to fully leverage cross-channel interactions and limiting their MCI effectiveness. In this paper, we present ChA-MAEViT, an MAE-based method that enhances feature learning across MCI channels via four key strategies: (1) dynamic channel-patch masking, which compels the model to reconstruct missing channels in addition to masked patches, thereby enhancing cross-channel dependencies and improving robustness to varying channel configurations; (2) memory tokens, which serve as long-term memory aids to promote information sharing across channels, addressing the challenges of reconstructing structurally diverse channels; (3) hybrid token fusion module, which merges fine-grained patch tokens with a global class token to capture richer representations; and (4) Channel-Aware Decoder, a lightweight decoder utilizes channel tokens to effectively reconstruct image patches. Experiments on satellite and microscopy datasets, CHAMMI, JUMP-CP, and So2Sat, show that ChA-MAEViT significantly outperforms state-of-the-art MCI-ViTs by 3.0-21.5%, highlighting the importance of cross-channel interactions in MCI. Our code is publicly available at https://github.com/chaudatascience/cha_mae_vit.

## 1 Introduction

Visual encoders generally process fixed-channel inputs, such as RGB, during both training and testing phases [2–9]. However, in satellite imaging [10], robotic sensing [11], cell microscopy [12, 13], and medical imaging [14], the input data can vary in both the number and type of channels. These varying channel configurations arise from differences in sensor modalities, acquisition settings, or experimental conditions. Multi-Channel Imaging (MCI) models are designed to learn feature representations from heterogeneous channels whose type and number vary during both training and testing [12, 13]. This adaptability allows a single model to effectively support diverse channel configurations, thereby reducing computational costs and minimizing the risk of overfitting [12].

Prior MCI-Masked Autoencoder (MAE) models (*e.g.*, [1, 15]) demonstrated promise in capturing local spatial structures by learning to reconstruct randomly masked patches in multi-channel images. These approaches assume that images exhibit significant redundancy across channels, enabling the reconstruction of masked patches using unmasked patches from similar channels. While this

39th Conference on Neural Information Processing Systems (NeurIPS 2025).

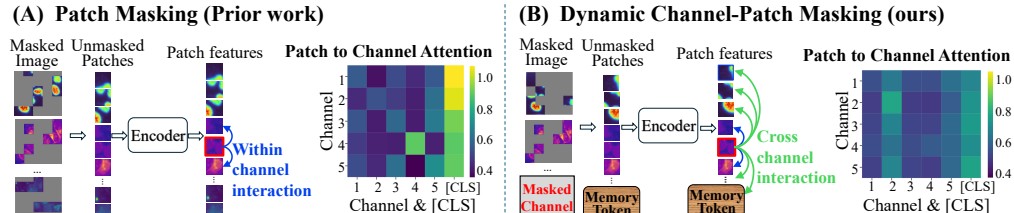

Figure 1: **Image patch interactions in MCI. (A) Prior Work on MCI-MAEs** (*e.g.*, CA-MAE [1]) employ random patch masking to train the model to reconstruct the masked patches. In the attention map, where each row represents the average attention score of all patches in a channel towards other channels and the [CLS] token (last column), we show each patch primarily attends to its own channel (the diagonal) and the [CLS] token. This suggests that patch masking may not effectively promote cross-channel interactions in MCI. **(B) In contrast, Dynamic Channel-Patch Masking (ours)** encourages more interactions between patches across different channels by using both channel and patch masking. Also, *memory tokens* serve as long-term memory to help information aggregation across channels. Its attention pattern demonstrates a more uniform distribution across channels, indicating that each image patch can learn more meaningful interactions.

assumption works well for natural images, where color channels typically show strong correlations, it presents challenges in MCI scenarios. In these situations, complementary sensors (*e.g.*, multispectral and LiDAR) may capture distinct physical properties with minimal feature overlap. As shown in Fig. 1(a), when using patch masking, the patch-to-channel attention concentrates on its own channel (the diagonal) and the [CLS] token. This indicates that prior MAE methods mainly focus on visible patches within the same channel, neglecting cross-channel feature interactions, restricting the model's ability to learn useful features that require information from multiple channels.

To address this challenge, we introduce **Ch**annel-**A**ware **MAE** − multichannel **Vi**sion **T**ransformer (ChA-MAEViT), which enhances cross-channel learning in MCI through four key improvements summarized in Fig. 2. First, we propose Dynamic Channel-Patch (DCP) Masking, which adaptively masks both patches and channels during training, compelling the model to reconstruct these missing patches and channels using the remaining patches (Fig. 2, left). DCP Masking adjusts the channel masking ratio to enhance feature learning and robustness to missing channels during inference. Fig. 1(b) shows our approach evenly redistributes patch attention scores across channels, highlighting improved cross-channel interactions. However, in MCI, reconstructing masked channels is challenging since each channel may encode unique features that are not easily inferred from others.

To enable the model to retain global information across all channels during reconstruction, we introduce the use of *memory tokens* in MCI (Fig. 2, middle). Inspired by register tokens [16], which are extra tokens added in the input to reduce artifacts in the feature maps of ViTs, these learnable embeddings serve as long-term memory to retain key cross-channel information in both the encoder and decoder. In addition, we use a hybrid token fusion module in the encoder to combine fine-grained patch tokens with the global class token for a richer representation (Fig. 2, middle).

Finally, we employ Channel-Aware Decoder that simultaneously processes patch tokens from all channels. Existing methods often rely on separate decoders for each channel [1, 17, 18], which do not scale well with many input channels. Our shared lightweight decoder utilizes channel tokens to provide channel-specific behavior and memory tokens to improve cross-channel feature reconstruction, enhancing performance while reducing computational costs.

Most of our experiments use both self-supervised (SSL) and supervised learning to boost performance, showing a similar complementary benefit from combining them (*e.g.*, [19–22]). However, Appendix F also evaluates SSL by itself, where we show our approach still outperforms prior work.

The most relevant work to ours focuses on enhancing cross-channel interactions [23–26]. However, these approaches mainly use single-modality images, such as RGB, where channel similarities facilitate strong correlations. As noted earlier, this often does not generalize to the more complex MCI domain where channels convey varying information, *e.g.*, a robot sensor can contain LiDAR, RGB, and thermal camera. Thus, MCI requires a balance between preserving unique information in each channel and effectively modeling the complex relationships between channels. MCI models must also be robust to missing channels to support varying channel configurations [12, 13]. This means that cross-channel interaction should not only improve the overall representation but also

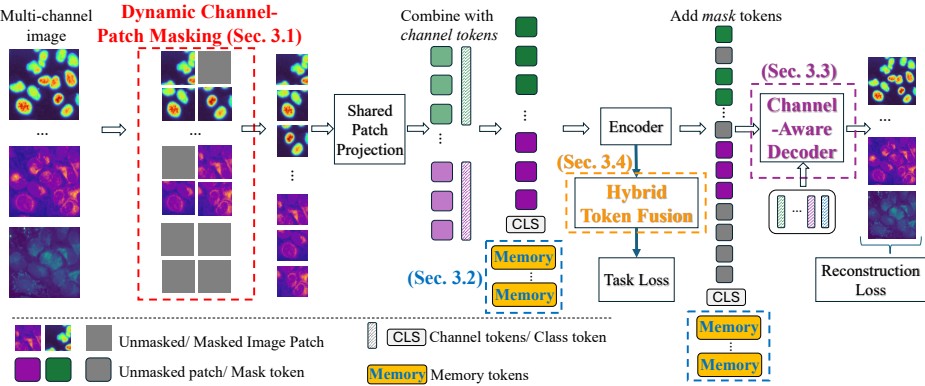

Figure 2: Our **ChA-MAEViT** approach enhances cross-channel learning via four key components: 1) **Dynamic Channel-Patch Masking**, which compels the model to reconstruct varying proportions of missing channels and patches, thus improving interactions across channels and robustness to the absence of some channels (Section 3.1). 2) **Memory Tokens**, which act as long-term memory to facilitate information sharing between channels (Section 3.2). 3) To reconstruct the masked patches and channels, we use a **Channel-Aware Decoder** that leverages channel tokens for image reconstruction, enhancing performance while minimizing computational costs (Section 3.3). 4) A **Hybrid Token Fusion** module, which combines fine-grained patch tokens with a global [CLS] token to improve feature representation (Section 3.4).

involve learning useful redundancy across channels. ChA-MAEViT addresses these challenges by introducing DCP Masking to enhance cross-channel interaction while allowing the model to learn important channel-specific features, resulting in a more robust and higher-performing model.

Our key contributions are as follows:
- We explore ChA-MAEViT, which integrates MAEs into MCI-ViTs for improved robustness that reports a gain of $3.0 - 21.5\%$ across three diverse MCI datasets, CHAMMI [12], JUMP-CP [27], So2Sat [10], and consistent gains on Cloud-38 [28].
- We propose Dynamic Channel-Patch Masking, which adaptively masks both channels and patches during training, requiring the model to reconstruct them to boost cross-channel interaction.
- We introduce a Channel-Aware Decoder that uses channel tokens to reconstruct image patches more efficiently, thereby enhancing performance for downstream tasks.
- We use *memory tokens* − learnable vectors to retain cross-channel information and aggregate channel-specific features, and *hybrid token fusion module* to merge the [CLS] token with fine-grained patch tokens for enhanced feature representation.

## 2 Related Work

Many MCI methods do not use self-supervised objectives [13, 29, 30], but rather focus on aspects like diversity and robustness to missing channels. However, as noted in the introduction, self-supervised objectives like Masked Autoencoders (MAEs) can provide complementary information to further boost performance. MAEs themselves are often built for RGB images (*e.g.*, [19, 31, 32]) or those that are also strongly aligned like thermal images [18], based on assumptions of strong correlations between channels does not extend to many MCI images. Many methods in other domains followed suit, performing channel-wise masking on protein markers [33] or masked both of spatial and spectral channels [34, 35]. However, these methods rely on a fixed ratio of masked channels, which can be inadequate for the varying channel configurations found in MCI. In addition, prior work uses dual-branch architectures that treats spatial and channel representations independently, limiting meaningful interactions between channels, especially when spatial alignment is not guaranteed. In contrast, we dynamically mask a variable number of channels each time. By combining both channel and patch-level masking within a single encoder, we achieve better feature interaction.

The work most similar to ours is CA-MAE [1], which randomly masks a fixed proportion of patches from all channels and employs distinct decoders for reconstruction. In contrast, ChA-MAEViT employs both channel and patch-level masking with a shared decoder, and memory tokens serve as long-term memory to facilitate feature interactions.

There are other types of self-supervised learning methods beyond MAEs that have been explored in other settings (*e.g.*, [31, 32, 36–45]). However, similar to MAEs, they often make assumptions that do not generalize to MCI images. For example, self-distillation methods (*e.g.*, SimSiam [40], BYOL [41], DINO [42]) process two different views through two encoders, and then map one view to the other using a predictor network. These methods often rely on complex augmentations designed for natural images to generate dual views, such as color jittering and Gaussian blur. These are less effective for MCI, where heterogeneous channels (*e.g.*, LiDAR, thermal, RGB) carry distinct physical properties. Such augmentations can distort critical features like depth or intensity, hindering the model's ability to capture cross-channel dependencies essential for MCI [46, 47]. This makes masked modeling techniques attractive (*e.g.*, MAE [31], SimMim [32]), as they do not rely on augmentations that may be challenging to apply with more complex and varied images.

## 3   Channel-Aware MAE for Improved Cross-Channel Learning

Given a multi-channel image (MCI) denoted as $X$, which consists of various channels $c_i \in C$, our objective is to train a model $M$ that takes $X$ as input to generate representations and/or predictions. Following [12, 13, 29, 30], we focus on the MCI setting where the model $M$ is trained using all available channels $C$ but tested with a subset of those channels $C_{\text{test}} \subseteq C$. We describe the details of the four main components of our proposed framework, which is illustrated in Fig. 2.

### 3.1   Dynamic Channel-Patch Masking

As shown in Fig. 1(a), image patches in prior work seem to attend mostly to their own channels while neglecting the others, potentially missing rich interactions among channels. To mitigate this issue, we propose Dynamic Channel-Patch (DCP) Masking, which integrates both patch and channel-level masking strategies. We start creating image patches (Appendix B), which results in $n$ tokens per channel, each with $d$-dimensional embeddings, for $c$ channels.

Our masking strategy consists of two components: *random patch masking* and *dynamic channel masking*. First, *random patch masking*, aims to generate a mask $\mathbf{p\_mask} \in \{0,1\}^{n \times c}$ that applies a fixed masking ratio $r_p$ (*e.g.*, 75%) to mask patches independently across each channel. Specifically, for each channel $j$, we uniformly sample a set of $\lfloor n \cdot r_p \rfloor$ positions from the $n$ available patch positions, denoted $S_j \subset \{1, 2, \ldots, n\}$ with $|S_j| = \lfloor n \cdot r_p \rfloor$. We then mask these patches as $\mathbf{p\_mask}_{i,j} = \begin{cases} 1 & \text{if } i \in S_j \\ 0 & \text{otherwise} \end{cases}$, where $\mathbf{p\_mask}_{i,j} = 1$ indicates the patch at position $i$ of channel $j$ is masked, and $\mathbf{p\_mask}_{i,j} = 0$ indicates unmasked. All channels have the same number of masked patches, but locations vary per channel because they are independently sampled, as opposed to prior work that first generates the mask for one channel and then replicates it to all other channels [35, 48].

The second component, *dynamic channel masking*, aims to generate a mask $\mathbf{c\_mask} \in \{0,1\}^{n \times c}$ that adaptively mask out some channels. Here, the term "dynamic" refers to the varying number of channels masked during training, rather than adaptation based on the specific input data. We start by uniformly sampling the number of channels to be masked, denoted as $k \sim \mathcal{U}\{0, 1, \ldots, c-1\}$. Then, we uniformly sample a set of $k$ channels, denoted as $\mathcal{C}' \subset \{1, \ldots, c\}$ with $|\mathcal{C}'| = k$, and mask these channels out as $\mathbf{c\_mask}_{\star,j} = \begin{cases} \mathbf{1}^n & \text{if } j \in \mathcal{C}' \\ \mathbf{0}^n & \text{otherwise} \end{cases}$. This masking approach is inspired by Hierarchical Channel Sampling (HCS) [30]. However, unlike HCS, which serves as a channel dropout technique that completely removes selected channels, our approach employs masked channels as supervisory signals, designating them as labels for the reconstruction process. This enables the model to directly learn inter-channel relationships, thus enhancing greater cross-channel feature interaction.

Finally, we integrate these two components for training images in our DCP Masking strategy by introducing hyperparameters $\text{p}_{\text{patch}}, \text{p}_{\text{channel}} \in (0,1)$. These values divide the unit interval into three sections to randomly determine how to use one or the other type of mask, as follows:

$$\mathbf{mask} = \begin{cases} \mathbf{p\_mask} & \text{if } s < \text{p}_{\text{patch}} \\ \mathbf{c\_mask} & \text{if } \text{p}_{\text{patch}} \leq s < \text{p}_{\text{patch}} + \text{p}_{\text{channel}} \\ \mathbf{p\_mask} \vee \mathbf{c\_mask} & \text{otherwise} \end{cases} \tag{1}$$

where $s$ is a selection value chosen uniformly at random. In practice, we found that optimizing the model with these two masking strategies simultaneously can be difficult, as it may lead to excessive information loss, making image reconstruction challenging. To address this, we adjust the hyperparameters to alternate between the two masking strategies separately. Specifically, setting $p_{patch} = p_{channel} = 0$ merges both patch and channel masks into a unified mask, and setting $p_{patch} = p_{channel} = 0.5$ allows the model to switch between patch and channel masks. We adopt these two straightforward configurations for all our experiments. Refer to Appendix D for the procedure of DCP Masking, and Appendix F.5 for more analysis on the hyper-parameters.

## 3.2 Memory Tokens

Reconstructing masked channels is challenging due to their inherent differences, since each channel encodes unique features not easily inferred from others, *e.g.*, reconstructing LIDAR from RGB. Inspired by the concept of register tokens [16], which are used to reduce artifacts in the feature maps of ViTs, we introduce *memory tokens* for MCI (Fig. 2, middle). These memory tokens are learnable embeddings that serve as long-term memory, allowing for the storage of global information across all channels. During training, these tokens gather channel-specific features using self-attention mechanisms, helping the model retain and propagate information across layers that might otherwise be lost due to masking. Additionally, these tokens assist in decoding image patches to facilitate the reconstruction process. During inference, memory tokens enable the model to retrieve stored context, effectively addressing the issue of missing channels. Similar to the [CLS] token, memory tokens are incorporated into the input during both training and inference. However, while the [CLS] token is utilized as the final representation, the memory tokens are excluded.

Formally, we prepend $l$ memory tokens $\{\mathbf{M_i}\}_{i \in [l]}$ into the input sequence, resulting in $[\mathbf{t}_{CLS}; \mathbf{M}_1; \ldots; \mathbf{M}_l; \mathbf{t}_{1,1}; \mathbf{t}_{2,1}; \ldots; \mathbf{t}_{n,c}]$, where $\mathbf{t}_{CLS}$ and $\mathbf{t}_{i,j}$ are the class token and patch token at $i$-th location of $j$-th channel, respectively. After masking some patches (Section 3.1), the remaining patches are fed into a transformer encoder. Following [29, 30], spatial information is incorporated through learnable positional embeddings, while channel-specific properties are captured by special *channel tokens*, each represented as a learnable embedding. These channel tokens are concatenated with the patch tokens and jointly processed by the transformer encoder and decoder.

## 3.3 Channel-Aware Decoder

After processing the unmasked tokens with the encoder, we reconstruct the masked patches using the unmasked ones with a Channel-Aware Decoder. Unlike prior MCI-MAE methods that use separate decoders for each channel or modality [1, 17, 18], we employ a single decoder to process tokens from all channels simultaneously (Fig. 2, right). This scales better to the number of input channels (up to 18 in our experiments), while also boosting performance.

Let the output sequence from the encoder be $[\hat{\mathbf{t}}_{CLS}; \hat{\mathbf{M}}_1; \ldots; \hat{\mathbf{M}}_l; \hat{\mathbf{t}}_1; \hat{\mathbf{t}}_2; \ldots; \hat{\mathbf{t}}_v]$, where $v$ denotes the number of visible (*i.e.*, unmasked) patches fed into the encoder. This sequence is shorter than the original image patch length due to the missing masked patches. To reconstruct these masked patches, we utilize $u = (n \cdot c - v)$ mask tokens $\mathbf{m}_i$, where each mask token is a shared, learned vector representing the missing patch.

**Incorporating Channel-Specific Information.** To reconstruct channel-specific information, we combine patch tokens, whether visible ($\hat{\mathbf{t}}_i$) or masked ($\mathbf{m}_i$), with their corresponding *channel tokens*, which are also optimized by the encoder. We define $f(\cdot)$ as a function that returns the corresponding *channel token* for a given patch token. The input to our Channel-Aware Decoder is thus:

$$\hat{\mathbf{T}} = [\hat{\mathbf{t}}_{CLS}; \hat{\mathbf{M}}_1; \ldots; \hat{\mathbf{M}}_l; \hat{\mathbf{t}}_1 + f(\hat{\mathbf{t}}_1); \hat{\mathbf{t}}_2 + f(\hat{\mathbf{t}}_2); \ldots; \hat{\mathbf{t}}_v + f(\hat{\mathbf{t}}_v); \mathbf{m}_1 + f(\mathbf{m}_1), \ldots, \mathbf{m}_u + f(\mathbf{m}_u)]$$

Incorporating channel tokens provides a more channel-specific context, which enhances the reconstruction process. Additionally, we incorporate learnable positional embeddings, which are shared with the encoder, to provide information about the position of each patch in the image.

**Patch Reconstruction.** We pass the token sequence $\hat{\mathbf{T}}$ into several Transformer Blocks, followed by a Decoder Head to reconstruct the pixels of each image patch. Let $\mathbf{T} \in \mathbb{R}^{(n \cdot c) \times d}$ represent the output of the patch tokens from the final Transformer Block. The Decoder Head is a linear layer $\mathbf{W}_{decoder} \in \mathbb{R}^{d \times p^2}$, where $p$ is the image patch size. The pixel reconstruction of the $i$-th patch of the

$j$-th channel, is computed as $\hat{\mathbf{x}}_{i,j} = (\mathbf{T}_{i,j} \cdot \mathbf{W}_{\text{decoder}}) \in \mathbb{R}^{p^2}$. We found that a lightweight decoder with just $1-2$ Transformer Blocks is sufficient, aligning with findings from prior work [19, 31].

**Reconstruction Loss.** We use standard $L2$ loss on the image pixel, and the Fourier space $L1$ loss introduced in [1], both computed only on the masked patches. Let $P = \sum_{i=1}^{n} \sum_{j=1}^{c} \mathbf{mask}_{i,j}$ be the number of masked patches of the input image. Given $\mathbf{x}_{i,j}, \hat{\mathbf{x}}_{i,j} \in \mathbb{R}^{p^2}$, the original pixels of patch $i$ in channel $j$, and its reconstruction, respectively, the reconstruction loss is as follows:

$\mathcal{L}_{\text{pixel}} = \frac{1}{P} \sum_{i=1}^{n} \sum_{j=1}^{c} \mathbf{mask}_{i,j} \cdot L_2(\mathbf{x}_{i,j}, \hat{\mathbf{x}}_{i,j}); \quad \mathcal{L}_{\text{fourier}} = \frac{1}{P} \sum_{i=1}^{n} \sum_{j=1}^{c} \mathbf{mask}_{i,j} \cdot L_1(|\mathcal{F}(\mathbf{x}_{i,j})|, |\mathcal{F}(\hat{\mathbf{x}}_{i,j})|)$

$$\mathcal{L}_{\text{recon}} = (1 - \lambda_f) \cdot \mathcal{L}_{\text{pixel}} + \lambda_f \cdot \mathcal{L}_{\text{fourier}}, \tag{2}$$

where $|\mathcal{F}|$ is the amplitudes of Fast Fourier Transform. Following [1], we set a fixed $\lambda_f$ to 0.01.

## 3.4 Hybrid Token Fusion

Prior work either utilizes the [CLS] token [16, 49] or average pooling of patch tokens [1, 19] for downstream tasks. We found that incorporating both in our framework yields better performance (Fig. 2, middle). Given the output sequence from the encoder, we use a learnable query vector $\mathbf{q}_{\text{patch}}$ to interact with the patch tokens. The query attends to the patch tokens through cross-attention, and then combines with the [CLS] token to generate a fused representation $f_{\text{fusion}} = \hat{t}_{\text{CLS}} \odot \sigma(\text{CrossAttention}(\mathbf{q}_{\text{patch}}, \mathbf{T}_p))$, where $\mathbf{T}_p = [\hat{\mathbf{t}}_1; \hat{\mathbf{t}}_2; \dots; \hat{\mathbf{t}}_v]$, $\odot$ represents the elementwise product, and $\sigma$ the sigmoid function. The fused representation, $f_{\text{fusion}}$, integrates global context from the [CLS] token while also being refined by detailed spatial information from the patch tokens. We then use a multi-layer perceptron (MLP) with GELU activation [50] to enhance the fusion representation and tailor it for downstream tasks: $f_{\text{final}} = \text{Linear}(\text{GELU}(\text{Linear}(f_{\text{fusion}})))$.

**Training Objective.** The output of Section 3.4 is then used to compute the task loss, *e.g.*, cross-entropy for classification. Our final loss consists of three components: the primary loss for the specific task $\mathcal{L}_{\text{task}}$, regularization term $\mathcal{L}_{\text{d}}$, and reconstruction loss $\mathcal{L}_{\text{recon}}$ (Eq. (2)) as follows:

$$\mathcal{L}_{\text{final}} = (1 - \lambda_{\text{recon}}) \cdot (\mathcal{L}_{\text{task}} + \lambda_{\text{d}} \cdot \mathcal{L}_{\text{d}}) + \lambda_{\text{recon}} \cdot \mathcal{L}_{\text{recon}}, \tag{3}$$

where $\lambda_{\text{d}}$ is to balance the task loss and the regularization term, and $\lambda_{\text{recon}}$ is to balance the reconstruction loss with other losses. For regularization $\mathcal{L}_{\text{d}}$, we use the losses introduced in [29], in which Channel Diversification Loss applies to the channel tokens, and Token Diversification Loss applies to the patch tokens to encourage diversity learning. We set a small $\lambda_{\text{d}}$ (*e.g.*, 0.001) as suggested by [29], and a fixed $\lambda_{\text{recon}} = 0.99$ for all experiments.

## 4 Experiments

**Datasets.** We evaluate on three cell microscopy and satellite datasets. *(i)* **CHAMMI** [12], a channel-adaptive benchmark consists of varying-channel images sourced from WTC-11, HPA and CP, with 3, 4, and 5 channels respectively. Together, these three datasets contain 220K microscopy images, in which, 100K images are for training, while the rest for testing across domain generalization tasks. *(ii)* **JUMP-CP** [27] is a cellular imaging dataset, where each image has 8 channels, consisting of 5 fluorescence and 3 brightfield channels. We use compound perturbation plate BR00116991, which contains 127K training, 45K validation, and 45K test images, across 161 classes. *(iii)* **So2Sat** [10] contains synthetic aperture radar and multispectral optical image patches from remote sensing satellites, with 18 channels (8 from Sentinel-1, 10 from Sentinel-2) and 17 climate zone classes. We use the city-split version, consisting of 352K training, 24K validation, and 24K test images.

**Baseline methods.** We compare to the following methods:
- **DepthwiseViT** [12] processes each input channel through a depthwise convolution layer, averages the filtered features, and feeds them into a ViT backbone.
- **TemplateMixingViT** [52–54] learns channel weights via shared parameter templates, forming a patch projection layer for a ViT backbone.
- **HyperNetViT** [51] uses a neural network to generate channel-specific weights, concatenating them into a patch projection layer for a ViT backbone.
- **CA-MAE** [1] extends MAE for multi-channel imaging. We add task loss to adapt this baseline.
- **ChAda-ViT** [13] employs a shared projection for channel-wise feature extraction, combining the tokens with positional and channel embeddings for ViT processing.

Table 1: **Comparison of channel adaptive models**. We report the mean accuracy with standard deviation on the test set of three runs. "Full" refers to inference on all channels, while "Partial" means inference on a subset of channels. ChA-MAEViT outperforms other baselines consistently across three datasets. Note: all models contain the same number of parameters, except CA-MAE due to its separate decoders (*e.g.*, $4X$ parameters on So2Sat).

| | CHAMMI [12] | JUMP-CP [27] | | So2Sat [10] | |
| --- | --- | --- | --- | --- | --- |
| Model | Avg score | Full | Partial | Full | Partial |
| HyperNetViT [51] | 56.08±0.41 | 47.07±0.47 | 42.43±0.65 | 60.73±0.24 | 41.88±0.85 |
| DepthwiseViT [12] | 61.80±0.43 | 49.86±0.45 | 44.98±0.71 | 60.41±0.22 | 43.41±1.10 |
| TemplateMixingViT [52–54] | 58.16±0.42 | 52.48±0.27 | 43.85±0.73 | 55.86±0.10 | 37.28±0.34 |
| CA-MAE [1] + Sup. loss | 59.15±0.28 | 69.54±0.12 | 20.93±0.25 | 64.21±0.41 | 15.75±0.83 |
| ChAda-ViT [13] | 63.93±0.42 | 65.03±0.98 | 42.15±2.33 | 56.98±0.46 | 12.38±2.03 |
| ChannelViT [30] | 64.97±0.58 | 67.51±0.35 | 56.49±0.53 | 61.03±0.17 | 46.16±0.40 |
| DiChaViT [29] | 69.77±0.44 | 69.19±0.47 | 57.98±0.41 | 63.36±0.11 | 47.76±0.23 |
| **ChA-MAEViT (ours)** | **74.63**±0.54 | **90.73**±0.14 | **68.05**±0.21 | **67.44**±0.38 | **52.11**±0.49 |

- **ChannelViT** [30] Similar to ChAda-ViT, while incorporating Hierarchical Channel Sampling.
- **DiChaViT** [29] improves ChannelViT with regularization terms and diverse channel sampling.

Additionally, we investigate the effect of combining the best supervised baseline, DiChaViT, with the following SSL to evaluate the importance of incorporating self-supervision: MAE [1, 31], SimCLR [37, 38], SimSiam [40], iBOT [43], and DINOv2 [44]. More details in Appendix C.

**Implementation details.** All baselines employ ViT-S (21M) as the backbone. In our method, we remove one transformer block from the encoder to accommodate the decoder, ensuring that our approach maintains the same number of parameters as the baselines. Refer to Appendix E for details.

**Metrics.** We reported top-1 classification accuracy for the classification tasks on So2Sat [10] and JUMP-CP [27]. For the representation learning tasks on CHAMMI [12], following [29], we report the average F1 scores across WTC-11 and HPA.

## 4.1 Results

Table 1 compares ChA-MAEViT with different MCI-ViTs. ChA-MAEViT significantly outperforms the baseline models by an average of $10.0\%$ across three datasets. Specifically, ChA-MAEViT exceeds the state-of-the-art model DiChaViT by $5.0\%$ on CHAMMI, a channel-adaptive benchmark. For JUMP-CP and So2Sat, we conducted evaluations in both *Full* (using all channels) and *Partial* (using subsets of channels) settings. In the full setting, our model surpasses the next best models by $21.5\%$ and $3.0\%$, respectively. Additionally, ChA-MAEViT demonstrates its robustness in the partial settings, where we test JUMP-CP using five fluorescence channels and So2Sat using eight Sentinel-1 channels, showing improvement of $10.0\%$ and $4.5\%$ points respectively over prior work.

Table 2 presents the impact of integrating various SSL methods with the top-performing supervised approach, DiChaViT [29]. In the last row, we train DiChaViT to reconstruct the masked patches using only our DCP Masking, without incorporating *memory tokens*, Hybrid Token Fusion, or Channel-Aware Decoder. While the combination with MAE gives the highest performance, DCP still significantly outperforms the best SSL-enhanced variant by $0.6-5.6\%$ across three datasets. The performance gap highlights the effectiveness of DCP, which substantially exceeds the improvements gained from solely combining with SSL objectives. Additionally, by only processing the unmasked patches, our method achieves significantly faster runtime compared to other SSL methods, *e.g.*, $6X$ faster than DINOv2, making it both performant and computationally efficient for MCI tasks. Refer to Appendix C and Appendix Fig. 6 for more discussion on runtime and FLOPs comparisons.

To further demonstrate our approach's ability to generalize, Table 3 reports segmentation performance on 38-Cloud [28], complementing our results on the classification (So2Sat [10], JUMP-CP [27]) and representation learning tasks (CHAMMI [12]) reported earlier. ChA-MAEViT outperforms all baselines across evaluation metrics, demonstrating its ability to boost multi-channel learning.

Table 2: **SSL methods when combined with the best supervised baseline DiChaViT [29]**. The last row shows the combination of DiChaViT with Dynamic Channel-Patch Masking. Incorporating SSL in DiChaViT results in improvements. Best result with our masking strategy. **Boldfaces** and underlines indicates best and second best numbers, respectively.

| DiChaViT [29] + | CHAMMI | JUMP-CP | | So2Sat | |
| --- | --- | --- | --- | --- | --- |
| | | Full | Partial | Full | Partial |
| SimCLR [37, 38] | 70.72±0.40 | 67.12±0.39 | 56.96±0.68 | 64.44±0.58 | 49.42±0.63 |
| SimSiam [40] | 70.44±0.38 | 68.64±0.56 | 57.72±0.69 | 64.07±0.31 | 48.52±0.71 |
| iBOT [43] | 70.71±0.44 | 68.87±0.51 | 57.81±0.47 | 63.11±0.32 | 47.84±0.54 |
| DINOv2 [44] | 70.03±0.28 | 66.91±0.52 | 56.18±0.66 | 63.42±0.27 | 49.20±0.72 |
| MAE [1, 31] | 70.27±0.78 | 78.62±0.78 | 64.21±0.73 | 62.88±0.49 | 47.76±0.62 |
| **DCP Masking** | **71.47**±0.53 | **84.02**±0.49 | **65.72**±0.43 | **66.02**±0.22 | **50.52**±0.45 |

Table 3: **Comparing segmentation Performance on 38-Cloud [28].** We report the average and standard deviation from three runs. ChA-MAEViT outperforms all baselines in all metrics.

| Model | Accuracy | IoU | Precision | Recall | F1 |
| --- | --- | --- | --- | --- | --- |
| ChannelViT [30] | 0.945±0.003 | 0.843±0.012 | 0.919±0.015 | 0.911±0.004 | 0.915±0.003 |
| DiChaViT [29] | 0.951±0.004 | 0.857±0.011 | 0.924±0.007 | 0.922±0.006 | 0.923±0.003 |
| CA-MAE [1] | 0.946±0.002 | 0.845±0.003 | 0.917±0.005 | 0.916±0.003 | 0.916±0.002 |
| DiChaViT [29] + CA-MAE [1] | 0.959±0.001 | 0.886±0.004 | 0.939±0.005 | 0.943±0.004 | 0.941±0.002 |
| **ChA-MAEViT (ours)** | **0.964**±0.001 | **0.894**±0.002 | **0.943**±0.002 | **0.945**±0.001 | **0.944**±0.001 |

Table 4: **ChA-MAEViT ablation study.** Replacing any component from ChA-MAEViT, such as substituting our *DCP Masking* with the random patch masking from CA-MAE [1] ("w/o DCP Masking"), results in a performance drop. The decline is most significant when *DCP Masking* is excluded. Incorporating all components consistently enhances performance across all three datasets.

| Model | CHAMMI [12] | JUMP-CP [27] | | So2Sat [10] | |
| --- | --- | --- | --- | --- | --- |
| | Avg score | Full | Partial | Full | Partial |
| **ChA-MAEViT** | **74.63** | **90.73** | **68.05** | **67.44** | **52.11** |
| w/o DCP Masking | 70.51 | 88.01 | 52.33 | 64.50 | 28.70 |
| w/o Hybrid Token Fusion | 73.84 | 88.25 | 66.23 | 65.48 | 51.40 |
| w/o Memory Tokens | 73.62 | 87.81 | 67.21 | 65.18 | 50.46 |
| w/o Channel-Aware Decoder | 72.95 | 87.52 | 67.05 | 65.78 | 49.88 |

**Ablation study of ChA-MAEViT.** Table 4 presents the model's performance when a component is replaced. Specifically, "w/o DCP Masking" indicates that we replace our DCP Masking with random patch masking in CA-MAE [1], "w/o Hybrid Token Fusion" uses [CLS] token instead of Hybrid Token Fusion module, "w/o Memory Tokens" means no memory token is being used, and "w/o Channel-Aware Decoder" indicates replacing our Channel-Aware Decoder with CA-MAE's Separate Decoders. The results highlight the critical role of each of the components, especially DCP Masking, as its removal has the most detrimental effect on performance.

**Inference under varying channel configurations.** Table 5 evaluates ChA-MAEViT and the best baseline DiChaViT [29] when tested on varying numbers of channels of JUMP-CP. We train the model using all eight channels and assess performance after removing channels, *e.g.*, testing with seven channels, as shown in column "7," we averaged all $C_8^7 = 8$ possible combinations (refer to Appendix Table 9 for detailed results). ChA-MAEViT demonstrates enhanced robustness, particularly when some channels are missing during inference.

**Comparing masking strategies.** Table 6 compares various masking strategies applied to ChA-MAEViT. For fixed ratio approaches, we test several ratios (*e.g.*, 50%, 75%) and report the best results. DCP Masking consistently outperforms others, achieving highest scores in both *Full* and *Partial* settings. Notably, while random patch masking is effective in *Full*, it experiences a significant performance drop in *Partial*. In contrast, dynamic channel masking methods adapted from [29, 30] provide better adaptability for partial settings, and our DCP Masking approach improves even more.

Table 5: **Performance on varying channel configurations during inference**. Columns report the *mean*(std) across all channel combinations on JUMP-CP [27], *e.g.*, "7" indicates testing on 7 out of 8 channels ($C_8^7 = 8$ combinations). ChA-MAEViT consistently shows improved robustness with missing channels at test time. Note that the reported std reflects variation across channel combinations, not model training variance. Refer to Table 9 in the Appendix for model variance report.

| | | | | Number of channels at inference | | | | |
| Method | 8 | 7 | 6 | 5 | 4 | 3 | 2 | 1 |
|---|---|---|---|---|---|---|---|---|
| DiChaViT [29] | 69.19 | 61.91(9.3) | 54.49(12.4) | 46.35(13.4) | 38.00(12.4) | 30.09(9.3) | 23.97(4.9) | 20.90(1.6) |
| **ChA-MAEViT** | **90.73** | **83.36**(8.3) | **74.55**(11.7) | **63.46**(13.8) | **50.85**(13.9) | **38.13**(11.2) | **27.62**(6.4) | **21.59**(2.1) |

Table 6: **Mask strategies in MCI when using with ChA-MAEViT**. While *Random Patch* and *Channel Sampling* perform similarly on Full channels, *Random Patch* experiences a significant drop in Partial channel settings. Dynamic Channel-Patch (DCP) Masking demonstrates its effectiveness in both Full and Partial channel settings, outperforming all strategies across three datasets.

| | CHAMMI | JUMP-CP | | So2Sat | |
| Mask Strategy | Avg score | Full | Partial | Full | Partial |
|---|---|---|---|---|---|
| Random Patch (fixed ratio) (*e.g.*, [1, 17, 19, 55, 56]) | 70.51 | 88.01 | 52.33 | 64.50 | 28.70 |
| Random Patch (dynamic) | 68.23 | 85.54 | 55.86 | 64.84 | 31.92 |
| Channel (fixed ratio) [18] | 47.97 | 73.00 | 65.25 | 65.22 | 38.59 |
| Hierarchical Channel Sampling (dynamic, adapted [30]) | 69.86 | 83.71 | 67.75 | 65.20 | 48.77 |
| Diverse Channel Sampling (dynamic, adapted [29]) | 71.57 | 84.69 | 67.86 | 65.49 | 51.05 |
| Channel + Patch (fixed ratio) [35, 48] | 48.46 | 69.41 | 62.19 | 62.20 | 32.18 |
| **DCP Combination ($p_{channel} = p_{patch} = 0$) (ours)** | 73.75 | 85.95 | **68.36** | **67.44** | **52.11** |
| **DCP Alternate ($p_{channel} = p_{patch} = 0.5$) (ours)** | **74.63** | **90.73** | 68.05 | 66.47 | 50.69 |

In addition, we analyze two variants of DCP that we use in all our experiments: Combination and Alternate. The DCP Combination unifies both patch and channel masks, while the DCP Alternate switches between the two for each training iteration. Both variants outperform traditional patch- and channel-based masking, highlighting the effectiveness of joint channel-patch masking. Refer to Appendix F.5 for more analysis and hyperparameter settings for the DCP Masking.

## 4.2 Model ablations and sensitivity analysis

**Impact of Memory tokens.** Fig. 3(a) & (b) show the accuracy achieved by numbers of *memory tokens*. Using memory tokens enhances performance, but beyond a certain point yields diminishing returns. This suggests that while memory tokens can improve performance, excessive reliance on them may negatively impact the interaction of patch features. We found that a default of 4 memory tokens works well across the three datasets.

**Attention patterns of Memory tokens.** Fig. 4 shows the attention patterns between image patches and memory tokens of the encoder. Each group of channels displays distinct preferences for specific memory tokens. Fig. 4(a) shows that the *VH* channels primarily focus on memory token 8, while the *Lee-filtered* channels have stronger attention toward memory token 1. Similarly, for JUMP-CP in Fig. 4(b), the *Brightfield* channels allocate more attention to memory token 3, whereas *Fluorescence* channels show a slight preference for memory token 1. This suggests that each type of channel utilizes different memory tokens to store the global information necessary for feature extraction.

**Token pooling strategies.** Table 7 compares the performance of various token pooling strategies on CHAMMI and So2Sat. In general, combining the [CLS] token with the average of the patch tokens gets better performance than either alone. By utilizing both the global [CLS] token and the fine-grained patch tokens, Hybrid Token Fusion consistently outperforms others across all settings.

**Reconstruction loss lambda.** Fig. 3(c) & (d) analyze the effect of the reconstruction weight $\lambda_{recon}$ (Eq. (3)) on model performance. Consistent with prior work [19, 22], a large value of $\mathcal{L}_{recon}$ gives the best results. However, performance drops significantly using just the MAE loss (*i.e.*, $\lambda_{recon} = 1$), noting the contributions of both losses. In all our experiments, we set a fixed $\mathcal{L}_{recon}$ to 0.99.

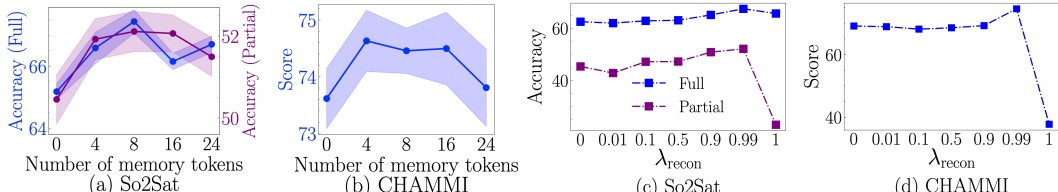

(a) So2Sat  (b) CHAMMI  (c) So2Sat  (d) CHAMMI

Figure 3: **Impact of the number memory tokens and reconstruction lambda $\lambda_{\text{recon}}$ (Eq. (3)). (a) & (b)** Using $4 - 8$ tokens improves performance, however, using more memory tokens (*e.g.*, 24) may reduce the effectiveness. **(c) & (d)** $\lambda_{\text{recon}} = 0$ means without the reconstruction loss, while $\lambda_{\text{recon}} = 1$ indicates only using the reconstruction loss. For $\lambda_{\text{recon}} = 1$ on So2Sat, we run linear probing. $\lambda_{\text{recon}} = 0.99$ works best for ChA-MAEViT on both datasets.

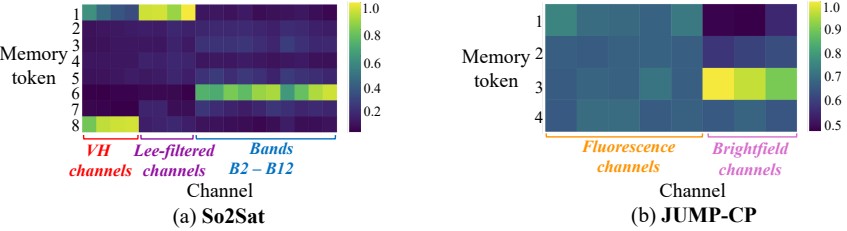

(a) **So2Sat**  (b) **JUMP-CP**

Figure 4: **Attention between image patches and memory tokens of the encoder**. Each channel group focuses on different memory tokens. **(a) So2Sat:** *VH* channels utilize memory token 8, whereas *Lee-filtered* channels attend more to memory token 1. **(b) JUMP-CP:** *Brightfield* channels focus on memory token 3, while Fluorescence channels favor memory token 1.

Table 7: **Comparison of token pooling methods.** Hybrid Token Fusion achieves the best performance, demonstrating the benefit of leveraging both global and fine-grained local tokens.

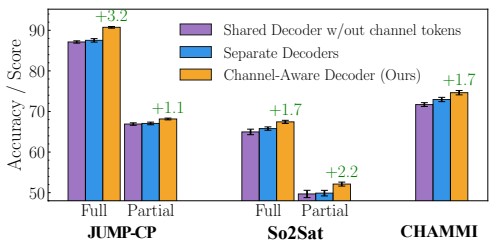

|  | CHAMMI | So2Sat | |
| --- | --- | --- | --- |
| Token Pooling Method | | Full | Partial |
| Avg Patch tokens | 71.41 | 64.86 | 51.87 |
| [CLS] token | 73.84 | 65.48 | 51.40 |
| [CLS] + Avg Patch tokens | 73.96 | 66.23 | 49.45 |
| **Hybrid Token Fusion (ours)** | **74.63** | **67.44** | **52.11** |

Figure 5: **Different Decoders when using with ChA-MAEViT**. Our Channel-Aware Decoder outperforms the best baseline by $1.1 - 3.2\%$ on all three datasets.

**Channel-Aware Decoder analysis.** Fig. 5 shows Channel-Aware Decoder in ChA-MAEViT outperforms both Separate Decoders [1] and Shared Decoder W/out Channel Tokens by $1.1 - 3.2\%$ across three datasets in full and partial settings. Channel-Aware Decoder is also more efficient than Separate Decoders thanks to its shared parameters, *e.g.*, $18X$ fewer parameters on So2Sat.

## 5 Conclusion

In this paper, we introduce ChA-MAEViT, a novel MAE-based method to enhance feature learning in Multi-Channel Imaging (MCI) ViTs. First, we introduce Dynamic Channel-Patch Masking, in which we adaptively mask both image patches and channels and train the model to reconstruct the masked patches. Additionally, we incorporate *memory tokens* to preserve global context across channels and Hybrid Token Fusion module to combine features from local patch tokens and global class tokens. Furthermore, we propose Channel-Aware Decoder to efficiently reconstruct channel-specific details. Experiments conducted on the CHAMMI, JUMP-CP, and So2Sat datasets demonstrate that ChA-MAEViT outperforms prior MCI-ViTs by $3.0 - 21.5\%$, highlighting the significance of enhanced cross-channel interactions. Future research could explore the adaptation of the channel-aware framework to additional complex modalities, such as volumetric medical imaging, which is composed of sequential two-dimensional slices.

## Acknowledgments and Disclosure of Funding

This study was supported, in part, by the National Science Foundation under NSF-DBI awards 2134695 and 2134696. Any opinions, findings, and conclusions or recommendations expressed in this material are those of the author(s) and do not necessarily reflect the views of the supporting agencies.

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

## A  Broader Impacts and limitations

The development of $\mathrm{ChA\text{-}MAEViT}$ represents a significant advancement in Multichannel Imaging, with benefits such as improved medical diagnostics and faster healthcare research. Its applications in satellite imaging also hold potential for environmental monitoring. However, there are risks, including the possibility of misuse for invasive surveillance systems, highlighting the need for ethical considerations and responsible deployment.

While our model is capable of handling unseen channels by leveraging relationships learned from known ones (Appendix F.10), we have not thoroughly evaluated its performance in such scenarios. Adapting to novel channels requires mapping them to existing ones, which becomes more challenging under domain shifts. We leave a systematic investigation of this setting to future work. Additionally, our approach requires some extra hyper-parameter tuning, which can increase computational resource demands.

## B  Image patching

Formally, let $I \in \mathbb{R}^{h \times w \times c}$ represent a multi-channel input image, where $(h, w)$ denotes the dimensions of the image and $c$ is the number of channels. We begin by patchifying each channel into $n$ 2-D patches of size $p \times p$ in pixels. This process yields $n \times c$ patches, denoted as $\mathbf{x}_{i,j} \in \mathbb{R}^{p \times p}$, where $n = \frac{h \cdot w}{p^2}$ is the number of patches per channel, $i$ indicates the spatial location of a given patch, and $j$ denotes the $j$-th channel. Next, we flatten each patch $\mathbf{x}_{i,j} \in \mathbb{R}^{p^2}$, and then pass it into a shared linear projection $\mathbf{W} \in \mathbb{R}^{p^2 \times d}$. This results in patch tokens $\mathbf{t}_{i,j} = (\mathbf{x}_{i,j} \cdot \mathbf{W}) \in \mathbb{R}^d$, leading to a sequence of $n \times c$ patch tokens $[\mathbf{t}_{1,1}; \mathbf{t}_{2,1}; \ldots; \mathbf{t}_{n,c}]$. Note that we utilize a shared projection across all channels, thus the number of parameters remains constant regardless of the number of input channels.

## C  Combination of DiChaViT with SSL

We adopt the following SSL methods with DiChaViT [29] as baselines: SimCLR [37, 38], Sim-Siam [40], MAE [31], iBOT [43], and DINOv2 [44].

SimCLR [37, 38] utilizes augmentation strategies that include random cropping, color distortion, and Gaussian blur to generate the dual views of an input image. This approach requires twice the computational resources per sample compared to single-path methods. Since hue and saturation are not well defined in multi-channel images, we apply the augmentation by only adjusting brightness and contrast.

SimSiam [40] employs similar augmentations as SimCLR, but it eliminates the use of negative pairs by stop-gradient. This operation helps prevent representation collapse by decoupling the optimization of the twin network branches, which simulates an alternating optimization process similar to the Expectation-Maximization (EM) approach. Additionally, SimSiam does not require momentum encoders.

MAE [31] involves randomly masking a large portion of the input image and training the model to reconstruct the missing content. Unlike contrastive methods, MAE does not depend on augmentation pipelines, which can be ineffective for heterogeneous channels in MCI [46, 47]. Additionally, by processing only a small fraction of the image, this method significantly lowers computational demands, thereby enhancing efficiency. Note that combining MAE with DiChaViT [29] results in a model very similar to CA-MAE [1], an extension of MAE for multi-channel imaging. However, key components from DiChaViT— such as diverse regularizers, channel tokens, and the diverse channel sampling strategy — are not present in CA-MAE, making the combined model more expressive for handling multi-channel imaging.

iBOT [43] combines masked image modeling with self-distillation by utilizing dual augmentation streams that include both masked and full views. This encourages the model to align representations across different views while reconstructing missing content. Particularly, iBOT uses clockwise masking introduced in BEiT [57], where a block of image patches is masked each time. To adapt this clockwise masking for multi-channel images, we generate a mask for each channel, ensuring that the masked blocks only cover the areas in the channel. Compared to MAE, iBOT introduces

additional computational overhead due to its use of teacher-student distillation. Furthermore, the dual-stream processing increases the per-sample training cost, making iBOT more computationally demanding than single-view methods like MAEs.

DINOv2 [44] employs multi-cropping alongside various augmentations, such as color jittering, Gaussian blur, and random solarization, as well as self-distillation techniques. Notably, DINOv2 also incorporates additional loss functions, such as iBOT loss [43] and KoLeo loss [58]. During training, DINOv2 generates multiple crops from each image (*e.g.*, 2 global views and 8 local views) which leads to significantly higher computational demands compared to other SSL methods. For example, training DINOv2 on JUMP-CP dataset using two GPUs requires approximately 6 days, whereas SimCLR takes around 2 days and MAE only requires 1 day. Refer to Fig. 6 for FLOPs counts.

To integrate these SSL methods with DiChaViT [29], we add the task loss and optimize it along with the SSL loss. Additionally, we explored another variant that included an additional branch utilizing standard augmentations recommended by the authors of the dataset to optimize the task loss and return the predictions at test time (*e.g.*, only using random cropping, horizontal flipping, and thin-plate-spline transformation [59] for CHAMMI). We observed that relying exclusively on complex augmentation pipelines, like those used in SimCLR, negatively impacts the performance of the combined model, while incorporating at least one view with regular augmentation enhances the model's learning capability. For self-distillation methods such as DINOv2, we found that the performance of the teacher network is slightly better than that of the student network, thus we report the performance on the teacher network.

## D    Dynamic Channel-Patch (DCP) Masking Algorithm

Algorithm 1 outlines the procedure of Dynamic Channel-Patch (DCP) Masking in Section 3.1 of the main paper, where we combine both *patch* and *channel* masking strategies. In practice, setting $p_{patch} = p_{channel} = 0$ merges both patch and channel masks into a unified mask, and setting $p_{patch} = p_{channel} = 0.5$ allows the model to switch between patch and channel masks. We adopt these two straightforward configurations for all our experiments. For $p_{patch} = p_{channel} = 0$ (*i.e.*, *DCP Combination*), we found that it works well with a small value of patch mask ratio $r_p$, and simply set $r_p = 0.25$ for all the experiments. For $p_{patch} = p_{channel} = 0.5$ (*i.e.*, *DCP Alternate*), we used a larger value of $r_p = 0.75$. Refer to Appendix F.5 for more analysis on these hyperparameters.

## E    Implementation details

For most of the baseline models, such as HyperNetViT, ChannelViT, ChAda-ViT, and DiChaViT, we utilize the implementation from [29] [1]. For iBOT and DINOv2, we adapt the implementation from [44] [2]. We employ a ViT small architecture with 21M parameters as the backbone for all methods. To ensure that we maintain the same number of parameters as the baselines, we removed one layer from the encoder to incorporate it into the decoder for our method. For both CHAMMI and JUMP-CP, we use a patch size of 16, while for the So2Sat dataset, we opt for a patch size of 8. We trained the models using the AdamW optimizer, aiming to minimize the cross-entropy loss for the JUMP-CP and So2Sat datasets, while employing proxy loss for CHAMMI.

**CHAMMI dataset [12].** For the baselines, we utilize the [CLS] token from the final layer as the feature representation and train the model to minimize the proxy loss [60]. We then evaluate the model on various tasks using the evaluation code provided by [12], incorporating a 1-Nearest Neighbour classifier to calculate the macro-average F1-score for each task individually [3]. The channel-adaptive interfaces are adapted from the authors' implementation code [4]. In addition to the model, we apply the same data augmentation techniques recommended by the authors, including thin-plate-spline (TPS) transformations [59]. Each model is trained for 100 epochs with a learning rate of 0.0004 and a batch size of 64.

---

[1]`https://github.com/chaudatascience/diverse_channel_vit`
[2]`https://github.com/facebookresearch/dinov2`
[3]`https://github.com/broadinstitute/MorphEm`
[4]`https://github.com/chaudatascience/channel_adaptive_models`

---

**Algorithm 1:** Dynamic Channel-Patch Masking

---

**Input** : Number of patches per channel $n$;
     Number of channels $c$;
     Patch mask ratio $r_p$;
     Probability of using patch masking $p_{patch}$;
     Probability of using channel masking $p_{channel}$;
      $(0 \leq p_{patch} + p_{channel} \leq 1)$

  // Random Patch Mask

1 **p_mask** $= \mathbf{0}^{n \times c}$  *// initialize patch mask*
2 **For** $j = 1, 2, \ldots, c$:
3  **p_mask**$[:, j] = \text{generate\_random\_patch\_mask}(n, r_p)$
4 **EndFor**

  // Dynamic Channel Mask

5 Uniformly sample a number of channels to mask $k \sim \mathcal{U}\{0, 1, \ldots, c-1\}$
6 Uniformly sample $k$ masked channels $\mathcal{C}' = \text{Sample}(\{1, \ldots, c\}, k)$
7 **c_mask** $= \mathbf{0}^{n \times c}$  *// initialize channel mask*
8 **For** $j \in \mathcal{C}'$:
9  **c_mask**$[:, j] = \mathbf{1}^n$  *// mask whole channel $j$*
10 **EndFor**

  // Masks For Current Iteration

11 Sample $s \sim \mathcal{U}(0, 1) \in \mathbb{R}$
12 **If** $s < p_{patch}$:
13  **mask** $= \mathbf{p\_mask}$ // Using patch mask
14 **Else If** $s < p_{patch} + p_{channel}$:
15  **mask** $= \mathbf{c\_mask}$ // Using channel mask
16 **Else**: // Combining both masks
17  **mask** $= \mathbf{p\_mask} \vee \mathbf{c\_mask}$

**Output** : **mask** $\in \{0, 1\}^{n \times c}$, where 1 denotes mask

---

**JUMP-CP [27] and So2Sat [10] datasets**. Following [29, 30], we warm up the learning rate for the first 10 epochs, reaching a peak of $0.0004$. After this initial period, the learning rate gradually decays to $10^{-6}$ according to a cosine scheduler. To mitigate overfitting, we apply a weight decay of $0.04$ to the weight parameters while excluding the bias and normalization terms. We use the same data augmentation techniques as outlined in [29]. For the final prediction, we utilize [CLS] token from the Transformer encoder into a classifier head to predict the probability of each class. The final model checkpoint is selected based on validation performance. Each model is trained for 100 epochs, with a batch size of $64$ on JUMP-CP and $128$ on So2Sat.

**Compute resources.** Experiments conducted on So2Sat and CHAMMI utilize one NVIDIA RTX GPU with 48GB RAM, alongside three Intel(R) Xeon(R) Gold 6226R CPUs. For JUMP-CP experiments, two NVIDIA RTX GPUs and six Intel(R) Xeon(R) Gold 6226R CPUs were used.

## F  Additional Results and Analyses

### F.1  FLOPs Count for Combination of DiChaViT with SSL

In Fig. 6, we present the FLOPs for each of the SSL methods when combined with DiChaViT, shown in Table 2 of the main paper. ChA-MAEViT achieves the lowest FLOPs, approximately one-sixth that of DINOv2.

### F.2  CHAMMI Benchmark Results

Table 8 presents the F1 scores of various MCI-ViT models (ViT-S backbone) evaluated on the CHAMMI benchmark [12]. The evaluation consists of nine tasks, including six out-of-distribution (OOD) tasks. ChA-MAEViT demonstrates superior overall performance, achieving the highest scores in six out of the nine tasks across different settings.

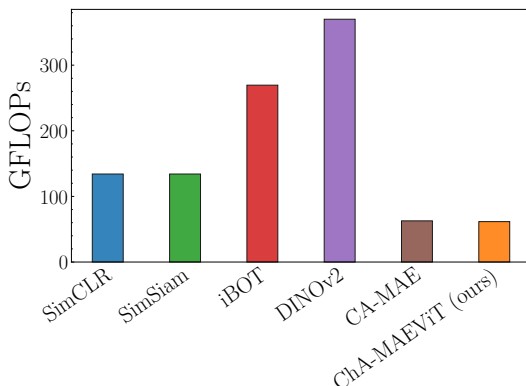

Figure 6: **FLOPs Count for Combination of DiChaViT with SSL**. We show FLOPs for each method when trained on JUMP-CP. ChA-MAEViT achieves the lowest FLOPs, $\approx 1/6$ that of DINOv2.

Table 8: **F1 Scores of MCI-ViT models on CHAMMI benchmark** [12]. ChA-MAEViT demonstrates better overall performance on CHAMMI, achieving the highest scores in 6 out of 9 tasks across various settings. "OOD" refers to out-of-distribution tasks. All models have the same number of parameters, except CA-MAE with $3X$ more parameters due to its use of separate decoders.

| | Average OOD | | | WTC | | HPA | | | CP | | | |
|---|---|---|---|---|---|---|---|---|---|---|---|---|
| Model | Mean | WTC | HPA | CP | Task1 | Task2 | Task1 | Task2 | Task3 | Task1 | Task2 | Task3 | Task4 |
| HyperNetViT [51] | 47.17 | 45.78 | 67.61 | 28.11 | 58.83 | 45.78 | 88.78 | 82.70 | 52.52 | 82.13 | 53.74 | 23.16 | 7.42 |
| DepthwiseViT [12] | 50.44 | 52.19 | 71.41 | 27.72 | 69.81 | 52.19 | 91.65 | **88.04** | 54.78 | 81.24 | 54.08 | **23.21** | 5.87 |
| TempMixingViT [52–54] | 47.33 | 51.52 | 64.80 | 25.66 | 61.66 | 51.52 | 85.01 | 79.91 | 49.69 | 77.45 | 48.83 | 22.56 | 5.60 |
| CA-MAE [1] + Sup. Loss | 48.82 | 61.85 | 56.45 | **28.15** | 77.57 | 61.85 | 84.45 | 71.89 | 41.01 | 76.49 | 56.52 | 19.43 | **8.49** |
| ChAda-ViT [13] | 50.82 | 67.18 | 60.67 | 24.60 | 77.58 | 67.18 | 87.49 | 75.94 | 45.41 | 83.92 | 45.58 | 21.94 | 6.28 |
| ChannelViT [30] | 52.54 | 67.58 | 62.35 | 27.81 | 78.36 | 67.58 | 83.93 | 76.73 | 47.97 | 77.70 | 55.16 | 21.89 | 6.38 |
| DiChaViT [29] | 55.36 | 75.18 | 64.36 | 26.53 | 80.87 | 75.18 | 88.08 | 79.26 | 49.45 | 84.08 | 53.03 | 20.95 | 5.60 |
| **ChA-MAEViT (ours)** | **58.02** | **77.15** | **72.11** | 24.81 | **84.52** | **77.15** | **94.14** | 87.47 | **56.75** | **90.89** | **56.68** | 10.25 | 7.50 |

## F.3 Performance when only using SSL Objectives

We evaluate the effectiveness of ChA-MAEViT in the SSL setting by comparing the performance of different SSL methods. Specifically, all models were trained solely using SSL objectives, *i.e.*, no task-specific loss was incorporated. For the So2Sat [10] and JUMP-CP [27] datasets, we reported the accuracy from linear probing over 20 epochs. As shown in Fig. 7, ChA-MAEViT achieves the highest score of $37.7\%$, surpassing other approaches by $1.0 - 8.5\%$ on CHAMMI. In Fig. 8, our method consistently outperforms baseline models in both Full and Partial settings on So2Sat. A similar trend is observed for JUMP-CP in Fig. 9. This demonstrates the effectiveness of our method in SSL settings for MCI, highlighting its ability to extract meaningful representations in scenarios where supervised labels are unavailable.

## F.4 Leave-One-Channel-Out at Test Time

In Table 9, we present the results of training the model using all eight channels of JUMP-CP and then testing it with various combinations of seven channels. This provides a detailed result for column "7" of Table 5 in the main paper, which represents $C_8^7 = 8$ different channel combinations. For each combination, we report the mean and standard deviation of the model's performance based on three runs. Our results demonstrate that ChA-MAEViT achieves an improvement of $17 - 23\%$ for each combination compared to the baseline models DiChaViT [29] and ChannelViT [30].

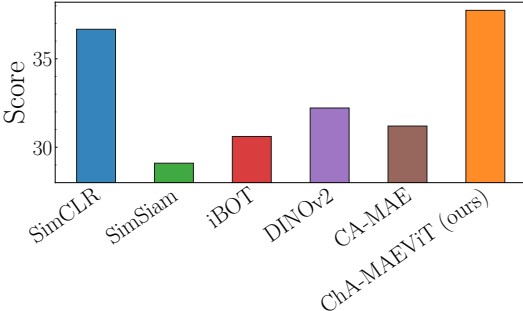

Figure 7: **Comparison of SSL methods on CHAMMI [12]**. All models were trained solely using SSL objectives, *i.e.*, without any task loss. ChA-MAEViT achieves the highest score, outperforming other baselines by $1.0 - 8.5\%$, demonstrating the effectiveness of our approach over other methods in SSL settings for MCI.

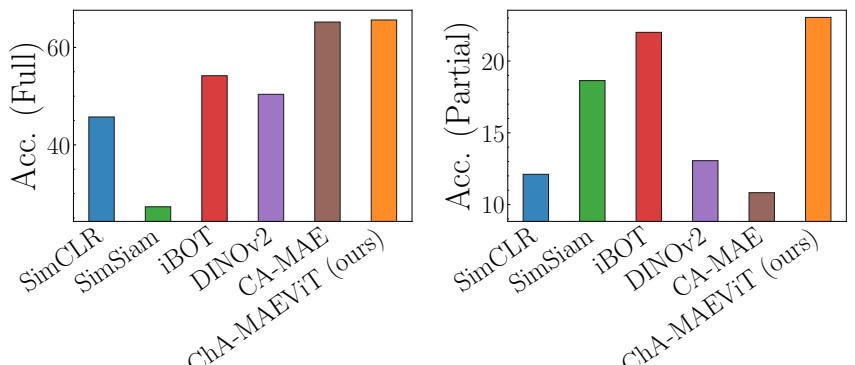

Figure 8: **Comparison of SSL methods on So2Sat [10]**. All models were trained solely using SSL objectives, *i.e.*, without any task loss, then trained with linear probing for another 20 epochs. ChA-MAEViT outperforms other baselines in both Full (left) and Partial channel settings (right), demonstrating its effectiveness for SSL in both scenarios.

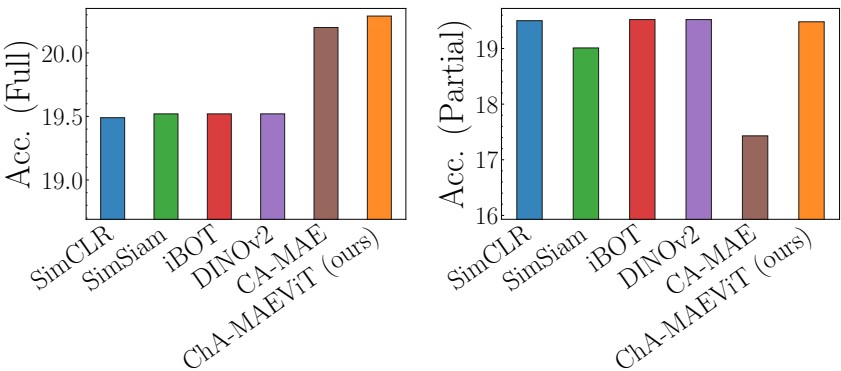

Figure 9: **Comparison of SSL methods on JUMP-CP [27]**. All models were trained solely using SSL objectives, *i.e.*, without any task loss, then trained with linear probing for another 20 epochs. ChA-MAEViT achieves the highest score when tested on Full channels (left) and performs comparably to the top-performing baselines on Partial channel settings (right).

Table 9: **Analysis of Leave-One-Channel-Out Testing.** We present the detailed results for column "7" in Table 5 of the main paper. Each row shows the results obtained by leaving out one channel and testing the model on the remaining seven channels. For all experiments conducted, we provide the *mean $\pm$ standard deviation* based on three runs. Our method, ChA-MAEViT, consistently outperforms the best two baselines ChannelViT [30] and DiChaViT [29] by $17-23\%$ across all tested combinations.

| Missing channel at inference | ChannelViT [30] | DiChaViT [29] | ChA-MAEViT (ours) |
|---|---|---|---|
| 0 | $61.72 \pm 0.48$ | $63.48 \pm 0.20$ | $\mathbf{86.29} \pm 0.37$ |
| 1 | $61.21 \pm 0.41$ | $62.72 \pm 0.28$ | $\mathbf{86.16} \pm 0.32$ |
| 2 | $61.90 \pm 0.48$ | $63.28 \pm 0.31$ | $\mathbf{87.13} \pm 0.24$ |
| 3 | $37.70 \pm 0.60$ | $38.83 \pm 0.46$ | $\mathbf{61.79} \pm 0.58$ |
| 4 | $58.52 \pm 0.63$ | $59.61 \pm 0.17$ | $\mathbf{82.67} \pm 0.53$ |
| 5 | $67.28 \pm 0.53$ | $69.12 \pm 0.16$ | $\mathbf{89.24} \pm 0.35$ |
| 6 | $67.20 \pm 0.59$ | $69.06 \pm 0.20$ | $\mathbf{87.54} \pm 0.20$ |
| 7 | $67.37 \pm 0.60$ | $69.21 \pm 0.19$ | $\mathbf{86.03} \pm 0.32$ |

## F.5 Dynamic Channel-Patch (DCP) Masking Analysis

*DCP Combination* **with varying masking ratios** $r_p$. Fig. 10 shows the impact of varying *random patch masking ratios* ($r_p$) on the performance of ChA-MAEViT when trained with DCP Combination ($p_{patch} = p_{channel} = 0$). The blue and purple lines indicate the performance for full and partial channels on JUMP-CP, respectively. As the masking ratio $r_p$ increases, accuracy for both full and partial channels decreases due to excessive information loss, making reconstruction more difficult. Since DCP Combination masks both patches and channels, it is reasonable to use a small $r_p$, in contrast to previous studies that often utilize a high ratio (*e.g.*, 0.75). For example, a $r_p$ of 0.25, when combined with dynamic channel masking, together mask $\approx 0.65$ patches in the original images. We found *DCP Combination* works well with a small value of $r_p$, thus we use $r_p$ of 0.25 as default value for all the datasets for *DCP Combination*.

*DCP Alternate* **with varying proportions of** $p_{channel}$ **and** $p_{patch}$. Fig. 11 evaluates the performance of *DCP Alternate* in ChA-MAEViT by varying the proportions of *channel-* and *patch*-level masking. We adjust $p_{channel}$, while setting $p_{patch} = 1 - p_{channel}$ to control the contribution of each masking strategy. For example, $p_{channel} = 1$ indicates exclusive channel-level masking, and $p_{channel} = 0$ denotes only using patch-level masking. The blue and purple lines represent accuracy on JUMP-CP for Full and Partial channel settings, respectively, across different $p_{channel}$ values. We observed that using both channel and patch masks together improves performance. Additionally, a higher channel-level masking proportion (*i.e.*, larger $p_{channel}$) improves accuracy in the Partial setting (purple) but leads to a decline in the Full setting (blue), highlighting the trade-off between these masking strategies. For *DCP Alternate*, we found that $r_p = 0.75$ works well across experiments.

**Patch Masking Strategies in DCP**. Fig. 12 compares two patch masking strategies with DCP. The first strategy, *Duplicate-Spatial Mask*, creates a patch mask for one channel and then duplicates the mask across all other channels (*e.g.*, [35, 48]). The second approach, *Independent-Spatial Mask*, generates a patch mask for each channel independently, resulting in varied patch positions. We set the same masking ratio for both strategies. Our findings indicate that using a different mask for each channel performs better than the duplicating one, yielding an improvement of $1.2-5.5\%$ across three datasets.

## F.6 Impact of memory tokens on reconstruction loss

Fig. 13 shows the effect of memory tokens on reconstruction loss during training. Evaluated on the So2Sat dataset, the model using memory tokens (blue line) achieves lower and smoother loss compared to those without them (red). This indicates that memory tokens may improve the model's ability to capture global information, helping to ease the learning process.

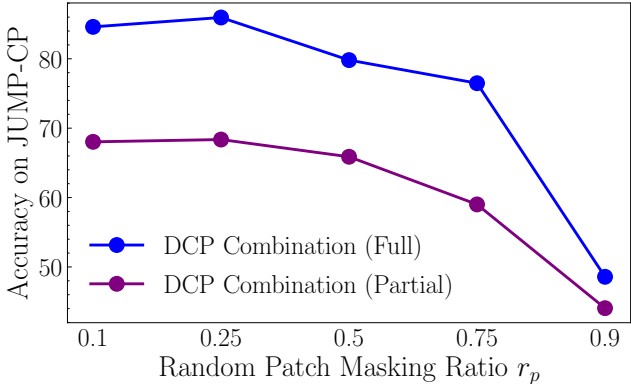

Figure 10: ***DCP Combination* with varying masking ratios** $r_p$. We train ChA-MAEViT using a combined mask integrating both channel- and patch-level masking, *i.e.*, *DCP Combination* ($p_{channel} = p_{patch} = 0$). The blue and purple lines depict accuracy on JUMP-CP for Full and Partial channel settings, respectively, across different patch masking ratios $r_p$.

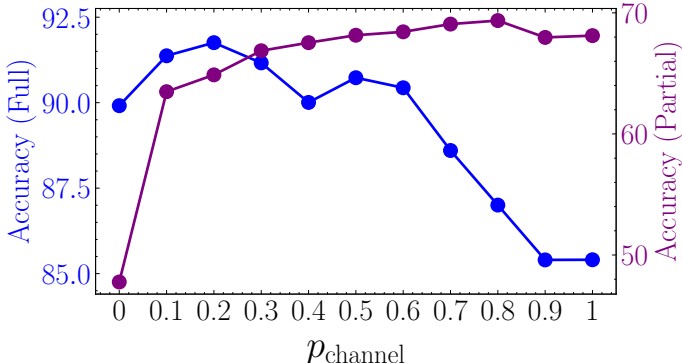

Figure 11: ***DCP Alternate* with different proportions of $p_{channel}$ and $p_{patch}$**. To evaluate the effects of *channel-* and *patch-* level masks in ChA-MAEViT during alternating masking, we vary $p_{channel}$ while setting $p_{patch} = 1 - p_{channel}$. $p_{channel} = 1$ indicates only using channel-level masking, whereas $p_{channel} = 0$ indicates only using patch-level masking. The blue and purple lines represent accuracy on JUMP-CP for Full and Partial channel settings respectively, across different values of $p_{channel}$.

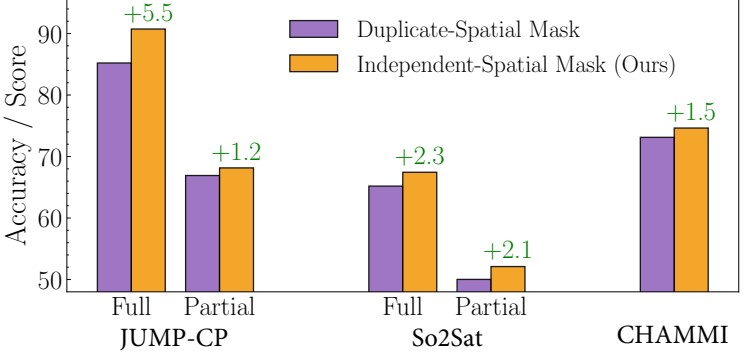

Figure 12: **Patch Masking Strategies in DCP**. We observed that using a different mask for each channel outperforms duplicating the same mask across all channels (purple), resulting in improvements of $1.2 - 5.5\%$ across three datasets.

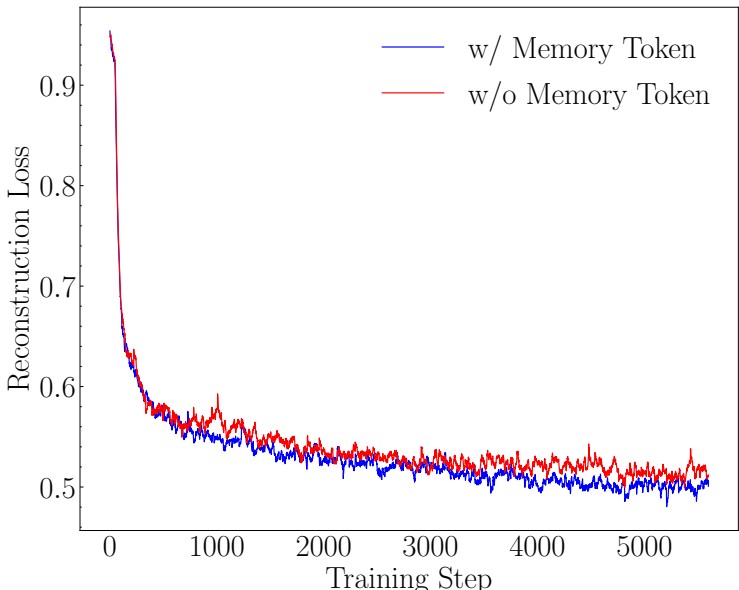

Figure 13: **Impact of memory tokens on the reconstruction loss**. Using memory tokens helps the reconstruction task, as demonstrated by the lower reconstruction loss observed on So2Sat.

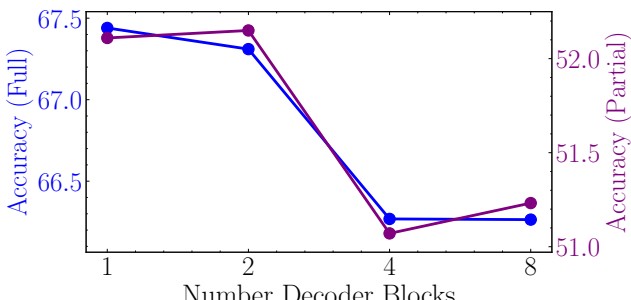

Figure 14: **Number of Decoder Blocks of ChA-MAEViT**. We show the performance of ChA-MAEViT when trained on So2Sat with different numbers of blocks in the decoder. We observed that using $1 - 2$ blocks gives the best performance.

### F.7 Number of Decoder Blocks of ChA-MAEViT

Fig. 14 illustrates the performance of ChA-MAEViT when trained on the So2Sat dataset with varying numbers of decoder blocks. We found that a configuration of 1 to 2 blocks yields the best performance.

### F.8 Scaling Model Size

Table 10 compares the performance of ChA-MAEViT with the two strongest supervised baselines: ChannelViT [30] and DiChaViT [29]. To evaluate the impact of model scaling, each model is evaluated using two backbone architectures: ViT-S (21M parameters) and ViT-B (85M parameters). Larger models exhibit improved performance, particularly on CHAMMI and JUMP-CP. Our approach consistently outperforms the baselines across all three datasets, regardless of the backbone used. In ChA-MAEViT, we reallocate one Transformer block from the encoder to the decoder, ensuring that all models maintain the same parameter count.

Table 10: **Scaling Model Size.** We compare ChA-MAEViT with the two best baseline models, ChannelViT [30] and DiChaViT [29]. Each model is evaluated using two backbones: ViT-S (21M parameters) and ViT-B (85M parameters). Increasing the model size boosts performance, particularly on CHAMMI and JUMP-CP. Our method outperforms others across three datasets with both backbones.

| Model | Backbone | CHAMMI [12] Avg score | JUMP-CP [27] Full | JUMP-CP [27] Partial | So2Sat [10] Full | So2Sat [10] Partial |
|---|---|---|---|---|---|---|
| ChannelViT [30] | ViT-S | 64.97 | 67.51 | 56.49 | 61.03 | 46.16 |
| DiChaViT [29] | ViT-S | 69.77 | 69.19 | 57.98 | 63.36 | 47.76 |
| **ChA-MAEViT (ours)** | ViT-S | 74.63 | 90.73 | 68.05 | 67.44 | 52.11 |
| ChannelViT [30] | ViT-B | 67.72 | 67.92 | 56.97 | 62.19 | 46.20 |
| DiChaViT [29] | ViT-B | 70.56 | 69.47 | 58.33 | 63.93 | 47.92 |
| **ChA-MAEViT (ours)** | ViT-B | **77.09** | **92.22** | **70.24** | **67.67** | **52.27** |

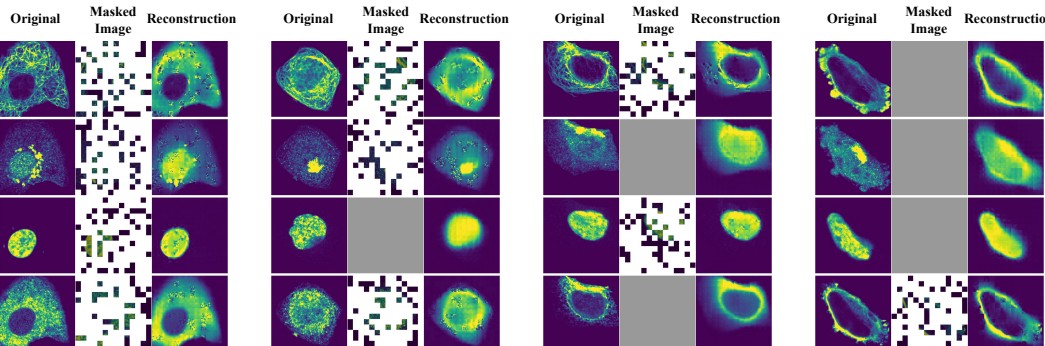

Figure 15: **Visualization of original, masked, and reconstructed images on CHAMMI-HPA [12], which contains 4 channels**. We show the reconstruction of ChA-MAEViT, utilizing a ViT-S/16 backbone. ChA-MAEViT demonstrates its capability to reconstruct masked channels and patches under different channel masking settings ($0 - 3$ masked channels with a fixed *patch masking ratio* $r_p = 75\%$). Gray blocks indicate fully masked channels.

### F.9    Reconstruction Images

Fig. 15 shows the *original*, *masked*, and *reconstructed* images from the CHAMMI-HPA dataset [12], which contains four channels. Each set of images shows the original image on the left, followed by the masked image created using Dynamic Channel-Patch Masking (with patch masking ratio $r_p = 75\%$), and finally, the reconstructed image produced by ChA-MAEViT (utilizing a ViT-S/16 backbone) on the right. The gray blocks indicate the entire channel has been masked. ChA-MAEViT demonstrates its ability to leverage cross-channel dependencies as it can reconstruct the entire channels using available patches from the other channels.

Note that the reconstructions lack some high-frequency details, as the primary goal of ChA-MAEViT is to learn robust, high-quality representations rather than producing detailed reconstructions. Our approach prioritizes the encoder's ability to capture high-level semantic and cross-modal relationships, with a deliberately lightweight decoder designed to validate the encoding of essential latent information. Additionally, following prior MAEs (*e.g.*, [1, 31, 61, 62]), we employ the mean squared error (MSE) loss for reconstruction, which often results in blurry outputs by averaging plausible solutions [62, 63]. This leads to reconstructions dominated by low-frequency components, an expected outcome of the design choice.

### F.10    Novel Channels at Inference

For the novel channels setting, we trained ChA-MAEViT using the first three channels (*i.e.*, channels $0, 1, 2$) and then tested it on two other channels (*i.e.*, channels $3, 4$) of JUMP-CP. One of the main challenges is that new channels do not have learned *channel tokens*. To address this, we assume that

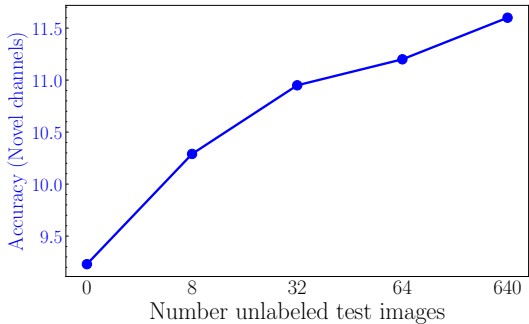

Figure 16: **Novel Channels at Inference**. Fine-tuning the *channel tokens* of novel channels with unlabeled test images boosted performance from $9.2\%$ (without fine-tuning, denoted as "0") to $11.6\%$ (with fine-tuning on 640 unlabeled test images).

some unlabeled test images are available at test time. We initialize the channel tokens for the new channels randomly and then fine-tuned these channel tokens on the unlabeled test images using the reconstruction loss in Eq. (2). Throughout this process, we kept the entire model frozen and only fine-tuned the newly initialized channel tokens.

In Fig. 16, we illustrate the accuracy on novel channels with varying numbers of unlabeled test images. For example, "0" indicates that no fine-tuning was performed, while "8" represents that eight test images (each containing only the two novel channels) were used to fine-tune the channel tokens before testing the model on the remaining test set. We observed that fine-tuning the channel tokens with some unlabeled test images improved performance from $9.2\%$ (without fine-tuning) to $11.6\%$ (with fine-tuning on 640 unlabeled test images). Additionally, to make an upper performance bound, we trained ChA-MAEViT on the training set of the novel channels (*i.e.*, channels 3 and 4). This model achieved an accuracy of $46.1\%$, indicating there is still a significant performance gap, which we leave for future work.

