# OpenReview forum: "ChA-MAEViT: Unifying Channel-Aware Masked Autoencoders and Multi-Channel Vision Transformers for Improved Cross-Channel Learning"
_NeurIPS.cc/2025/Conference — NeurIPS 2025 poster_

### Official Review · Reviewer_XYwZ · 2025-06-03

**Clarity:** 2
**Significance:** 2
**Originality:** 1
**Rating:** 2
**Confidence:** 5

**Summary:**

ChA-MAEViT proposes a masked autoencoding framework for Multi-Channel Imaging that alternates between masking patches and entire channels (DCP masking), uses a small set of memory tokens to capture global context, fuses [CLS] and patch embeddings via cross-attention, and decodes masked patches with channel-aware embeddings.

**Questions:**

1. Can the authors make their core contributions clearer, particularly with respect to prior existing work?

2. What is the performance drop if a known critical channel is masked?

3. How does the model handle truly unseen channel types at test time?

4. Does a frozen ChA-MAEViT encoder improve segmentation or regression compared to ViT or DiChaViT backbones?

**Ethical Concerns:**

["NO or VERY MINOR ethics concerns only"]

**Limitations:**

1. If some channels are rare during training, DCP masking may not adequately teach reconstruction, leading to overfitting to “full” channel sets.

2. Many hyperparameters (mask ratios, memory token count, reconstruction lambda) appear dataset-specific, reducing out-of-the-box usability.

**Quality:**

2

**Strengths And Weaknesses:**

1. Incremental novelty: All core ideas (channel masking, memory tokens, hybrid fusion, channel tokens) have direct antecedents (CA-MAE, DiChaViT, ViT register tokens).

2. Method is overly heuristic: Alternating between patch- and channel-masking lacks a principled basis; hyperparameters (ppatch, pchannel, mask ratios, reconstruction lambda etc.) require extensive tuning.

3. Limited task scope: Evaluated only on classification; no evidence it extends to segmentation or regression in MCI.

4. Unclear robustness: Experiments report averages over random channel removals, but worst‐case drops when critical channels are missing are not shown.

5. Unreported compute costs: No training/inference time comparisons versus simpler supervised baselines or existing MAEs.

---

> ### Author Rebuttal · Authors · 2025-07-31
>
> > Incremental novelty: All core ideas (channel masking, memory tokens, hybrid fusion, channel tokens) have direct antecedents (CA-MAE, DiChaViT, ViT register tokens).
>
> Thank you for the opportunity to clarify our contributions.
>
> First, **DCP Masking** is not derived from CA-MAE but is a novel method, as noted by reviewers **4uNy** and **La1T**. Unlike prior MAEs that assume cross-channel redundancy, DCP Masking addresses the low redundancy in multi-channel data by masking both channels and spatial patches to promote stronger cross-channel interactions. As shown in Tables 2 and 4, it outperforms CA-MAE's patch masking by 1-6% and 3-21% across three datasets with different backbones.
>
> Second, as stated throughout the paper, **memory tokens** and **Hybrid Fusion module** are adapted from existing ideas but are for novel purposes with a non-trivial shift in functionality: enhancing cross-channel interactions in masked modeling. Memory tokens capture global cross-channel semantics, while the Hybrid Fusion module leverages reconstructed patch tokens to enrich the final representation.
>
> Third, **Channel-Aware Decoder** introduces a scalable architecture that uses a shared lightweight decoder with channel tokens, unlike prior methods (e.g., CA-MAE) that require separate decoders per channel.
>
> Importantly, there is no guarantee combining prior ideas will work at all. Indeed, Table 2 shows merging the strongest baseline (DiChaViT) with CA-MAE yields only modest gains and still lags behind ChA-MAEViT by 3.5-12.5%. Our work investigates the role of all the components, both existing, newly proposed, and their integration, offering insights on what matters to solve open problems in the field.
> We’ll revise the camera-ready version to clarify these points.
>
> > Method is overly heuristic: Alternating between patch- and channel-masking lacks a principled basis;
>
> Alternating between masking strategies helps mitigate the learning difficulties caused by excessive information loss when channels and patches are masked simultaneously. As discussed in Appendix F.6 (line 652), a patch ratio of 0.25 when used with dynamic channel masking, results in about 65% of the spatial content being masked, comparable to the 75% masked in Alternate masking. Since each channel carries distinct semantics, masking both at once can be difficult to recover. By alternating the two, we observed more stable learning and up to a 5% performance gain on the JUMP-CP dataset (Table 4).
>
> > hyperparameters require extensive tuning and appear dataset-specific, reducing out-of-the-box usability.
>
> ChA-MAEViT does **not** require extensive or dataset-specific hyperparameter tuning. All our experiments used the same hyperparameters **without performing any separate tuning** for each dataset. Specifically:
>
> **Reconstruction Lambda**: fixed 0.99 for all experiments (line 308)
>
> **DCP Masking** has three hyperparameters (mask ratio, p_patch, p_channel). As detailed in Sec. 3.1 (line 156),  we used two fixed configurations: Alternate (0.75, 0, 0) and Combination (0.25, 0.5, 0.5), across all experiments. Table 4 shows both consistently outperform the best baseline in Table 1 by 2.5-21.0% and 3.5-16.5%, showing strong generalizability.
>
> **Memory Token**: As shown in Fig. 3, a small number of tokens (e.g., 4) boosts performance, with diminishing returns beyond that. Results are stable, showing low sensitivity to this hyperparameter.
>
> These results show that our **default settings are effective across datasets**, and can likely be used in new settings directly.
>
> We'd like to emphasize that additional hyperparameters do not necessarily diminish the method’s out-of-the-box usability, e.g., DINOv2 has over 20 and is widely adopted for its stability and performance. Ours are stable, transferable, and require no per-dataset tuning.
>
> To further support reproducibility and enhance understanding, we provide extensive analysis (Table 4, Fig. 3, and Appendix Figs. 9, 10, 12, 13), confirming ChA-MAEViT’s robustness and effective generalization across tasks.
>
> > Limited task scope: Evaluated only on classification; no evidence it extends to segmentation or regression in MCI.
>
> Only JUMP-CP and So2Sat are classification tasks. In CHAMMI, methods are evaluated based on their ability to learn useful cellular representations across various axes of generalization (e.g., new plates, new cell lines), using nearest neighbor protocols based on feature sets from the CHAMMI’s authors. Thus, CHAMMI assesses **representation learning, not a classification task**. ChA-MAEViT performs well on both tasks (3-21.5% boost across three datasets), showing a broad applicability that merits publication in its own right.  In addition, this is the same set of experimental settings of prior work [1,12,13,28], suggesting the community finds these two evaluation task types are effective.
>
> That said, to further demonstrate generalization, we evaluate segmentation performance on the 38-Cloud dataset [A], which includes four-channel (RGB + NIR) satellite images. Below are the average scores and std from three runs:
> |Model|Accuracy|IoU|Precision|Recall|F1|
> |-|-|-|-|-|-|
> |ChannelViT[13]|0.945±0.003|0.843±0.012|0.919±0.015|0.911±0.004|0.915±0.003|
> |DiChaViT[28]|0.951±0.004|0.857±0.011|0.924±0.007|0.922±0.006|0.923±0.003|
> |CA-MAE[1]|0.946±0.002|0.845±0.003|0.917±0.005|0.916±0.003|0.916±0.002|
> |DiChaViT[28]+CA-MAE[1]|0.959±0.001|0.886±0.004|0.939±0.005|0.943±0.004|0.941±0.002|
> |**ChA-MAEViT**(ours)|**0.964**±0.001|**0.894**±0.002|**0.943**±0.002|**0.945**±0.001|**0.944**±0.001|
>
> ChA-MAEViT outperforms all baselines across all evaluation metrics, demonstrating its effectiveness in enhancing multi-channel representation learning.
>
> [A] Cloud-Net: An End-To-End Cloud Detection Algorithm for Landsat 8 Imagery, Mohajerani et al., IGARSS’19
>
> > Unclear robustness: Experiments report averages over random channel removals, but worst‐case drops when critical channels are missing are not shown.
>
> Results in Table 1 follow established evaluation protocols from prior work, **not random channel removals**. CHAMMI’s varying channels stem naturally from differences in its three source datasets, i.e., we do not remove any channels. For So2Sat and JUMP-CP, we follow the established evaluation protocols of [2,3]. This can result in some critical channels never being masked, which is why we provided a more complete evaluation.
>
> Specifically, in Table 3, we report the average over **all possible channel combinations** as stated in its caption and L279, not random masking. E.g., column "4" shows the mean and std across all 70 four-channel combinations (8C4 = 70). Due to space constraints, only the mean and std values are shown in the main paper. However, detailed result in Appendix Table 8 **does show the worst-case drop**, when channel 3 is removed. Even in this worst-case setting, our method achieves the best result, outperforming the SOTA by 23%.
>
> To leave no further doubt, we extend Table 3 by including the minimum performance reported across all channel combinations below:
> |Method|8 channels|7 channels|6 channels|5 channels|4 channels|3 channels|2 channels|1 channel|
> |-|-|-|-|-|-|-|-|-|
> |DiChaViT[28]–avg|69.19±0.26|61.91±0.28|54.49±0.23|46.35±0.18|38.00±0.15|30.09±0.14|23.97±0.10|20.90±0.04|
> |ChA-MAEViT(ours)–avg|90.73±0.14|83.36±0.28|74.55±0.33|63.46±0.35|50.85±0.29|38.13±0.19|27.62±0.06|21.59±0.04|
> |DiChaViT[28]–min|69.19±0.26|38.80±0.45|29.39±0.25|23.49±0.21|20.42±0.26|19.76±0.03|19.75±0.02|19.73±0.01|
> |ChA-MAEViT(ours)–min|90.73±0.14|61.79±0.53|46.71±1.21|36.46±0.62|26.67±0.76|22.05±0.28|20.61±0.06|19.95±0.05|
>
> “avg” indicates the average across all possible combinations, while “min” represents the worst-case performance. Note that different from Table 3, here we report the average of three runs and the std for model variance.  As seen above, our approach consistently outperforms the prior state-of-the-art.
>
> > Unreported compute costs: No training/inference time comparisons versus simpler supervised baselines or existing MAEs.
>
> We report **training runtimes** in Appendix C (line 561), and **FLOPs comparisons** in Figure 5 of the Appendix. Specifically, on JUMP-CP, DINOv2 takes 6 days, SimCLR 2 days, and CAMAE/ChA-MAEViT 1 day. Figure 5 shows ChA-MAEViT has the lowest FLOPs, about 1/6 of DINOv2, while outperforming it.
>
> > How does the model handle truly unseen channel types at test time?
>
> In Appendix F.11, we show that ChA-MAEViT can handle novel channels unseen during training. We initialize new channel tokens for these channels and fine-tune only these tokens using a small set of unlabeled test images with DCP, while freezing the rest of the model. This approach improves performance as more unlabeled images are used (Fig. 16).
>
> > If some channels are rare during training, DCP masking may not adequately teach reconstruction, leading to overfitting to “full” channel sets.
>
> Rare channels do **not** lead to overfitting to full channel settings. DCP masking consistently uses a subset of channels during training (line 138), forcing the model to be robust to missing modalities and mitigating reliance on any specific channel configuration. Also, DCP masking outperforms patch masking by 4% (Table 4) on CHAMMI, in which some channels are always missing due to the nature of the dataset, meaning it never sees “full” channel sets.
> Rare channels, instead, reflect the natural data distribution and do **not** imply overfitting. We demonstrate robustness under these conditions in CHAMMI, where the “membrane” channel occurs only 31% as often as “nucleus.” Yet, our model outperforms the best baseline by 2% on images with the “membrane” channel (Table 7, Appendix), showing strong generalization under uneven channel distributions. Additionally, techniques like channel-aware sampling, oversampling, or importance-weighted loss can be easily integrated into our method to enhance learning if needed.

---

> > ### Author Response · Authors · 2025-08-05
> > **Official Comment by Authors**
> >
> > Dear Reviewer XYwZ,
> >
> > As we approach the end of the discussion period, we would greatly appreciate it if you could take a moment to read through our rebuttal. If you have any questions or need further clarification, we'd be happy to answer them. Thank you!
> >
> > -Authors of paper 17675

---

> > > ### Author Response · Authors · 2025-08-07
> > > **Official Comment by Authors**
> > >
> > > Dear Reviewer XYwZ,
> > >
> > > With about a day left in the discussion period, we wanted to kindly remind you to take a look at our rebuttal. We really appreciate your feedback and are happy to clarify or answer any questions you might have.
> > >
> > > -Authors of paper 17675

---

> > ### Comment · Reviewer_XYwZ · 2025-08-08
> >
> > I thank the authors for their rebuttal, which I have read in detail, including responses to my fellow reviewers' concerns.
> >
> > Despite these clarifications and additions, my core concerns regarding the paper's limited conceptual novelty and the heuristic nature of the proposed method remain. The rebuttal in fact reinforces the view that this work is a piece of engineering that combines existing concepts, rather than a fundamental contribution with a clear, principled foundation.
> >
> > ## 1. Novelty and conceptual contribution
> >
> > The authors argue for the novelty of DCP Masking and the unique purpose of their adapted components. However, the idea of masking or dropping entire channels for robustness is established in prior work, including ChannelViT and DiChaViT, which the authors build upon. DCP Masking is a new, more complex variant of this established idea, but it is not a fundamental conceptual leap. Alternating between channel- and patch-masking is a schedule choice rather than a new objective. Moreover, similar dual-masking schedules have been explored in audio-visual MAEs and in CA-MAE’s appendix.
> >
> > The authors' main argument is that a naive combination of DiChaViT + CA-MAE underperforms their method. Sure, this demonstrates they have found a specific integration that works. However, this result also frames the contribution as finding the right way to combine existing tools, which is strong engineering but does not resolve the issue of limited conceptual originality. The work primarily shows that this specific combination is effective, not why it represents a new fundamental principle of cross-channel learning.
> >
> > ---
> >
> > ## 2. Heuristic design & hyperparameter generality
> >
> > The authors argue that no dataset-specific tuning was done, citing fixed \lambda, two preset mask configurations and four memory tokens. Yet:
> >
> > i) The success of DCP still hinges on choosing when to alternate and on two masking ratios. The justification for alternating between patch and channel masking isto "mitigate learning difficulties caused by excessive information loss." This is a description of a practical workaround, not a principled design choice. It addresses a stability issue introduced by the method itself.
> >
> > ii) In general, the authors' defense of their hyperparameters is that the two tested configurations work well. However,
> >
> > a) Figure 3 shows non-monotonic behaviour w.r.t. token count. This suggests practitioners would need validation sweeps when moving to other channel counts.
> >
> > b) Appendix Figure 10 reveals a critical trade-off: increasing the channel masking proportion (p_channel) improves performance on partial-channel settings but degrades it on full-channel settings.
> >
> > These findings directly contradict the notion of a simple, "out-of-the-box" solution. It implies that a user must choose a configuration based on their expected inference scenario, reinforcing the critique that the method's complexity requires dataset- or task-specific tuning.
> >
> > The comparison to DINOv2 is moot, as no one is training DINOv2 from scratch. Its wide usability in the community is almost entirely due to the strength of the pretrained backbones that have been made publicly available, and through extensive testing have demonstrated wide in-the-wild visual generalization and feature utility.

---

> > > ### Author Response · Authors · 2025-08-09
> > > **Official Comment by Authors**
> > >
> > > Thank you for your response and for agreeing that our approach contains novel components and applications, even if we differ on two points, discussed below.
> > >
> > > **1. Strength of the contribution**
> > >
> > > This is inherently a subjective argument, especially in cases like this where there is general agreement that the components are different from prior work.  Reviewers **4uNy** and **La1T** agree with us that we provide a strong contribution, whereas the reviewer’s argument that our DCP Masking effectively reduces down to applying a masking schedule that has been studied in prior work is not well supported.  Specifically, we can find no mention of dual masking schedules in the CA-MAE appendix. Regardless, they only apply patch masking, whereas ours also leverages channel masking to improve the robustness to missing channels - a key difference in MCI methods from prior work. The reviewer did not provide a reference to audio-visual work to discuss, but our own search arrived at [A]. However, as it performs random masking on each modality, it is far more similar to CA-MAE than our approach. Specifically, it does not mask entire modalities to make the method more effective in the presence of only a single modality. Further, we actually compare to CA-MAE directly and find that our approach outperforms CA-MAE's patch masking by 1-6% and 3-21% with different backbones, and outperforms prior work by 3-21.5% across datasets (Tables 2 and 4). These results help demonstrate that our changes are effective.
> > >
> > > Now, even if we were to accept the reviewer’s view that there the DCP Masking and our other components with *memory tokens*, *hybrid fusion*, and a *channel-aware decoder* all simply amount to engineering efforts, we would highlight that there is a long history of high impact papers with similar contributions as our work, e.g., ViT, YOLOv2, ResNet, CLIP all could face similar criticisms.  For example, ViT “simply” splits up an image into patches so it fits into the expected input for a Transformer. These methods can be clear in hindsight, but creating the approach is far more challenging. A key question that would arise, if our approach is well-known and studied, why didn’t the authors of CA-MAE already use it?  The most likely explanation is that they didn’t consider it at the time, yet [A] was already published, i.e., if it was so obvious that our approach should do better and was known, then the authors of CA-MAE would certainly have done so.
> > >
> > > [A] Audiovisual Masked Autoencoders, Georgescu et al., ICCV’23
> > >
> > >  **2. Hyperparameter Generality Across Datasets and Tasks**
> > >
> > > We demonstrate strong out-of-the-box usability by using the same hyperparameters (i.e., two masking configurations while keeping other hyperparameters fixed), without any dataset-specific tuning across four diverse datasets on three task types (including an additional segmentation dataset the reviewer requested in the rebuttal). As shown in our paper, these two settings consistently yield improvements of 2.5-21.0% and 3.5-16.5% respectively over prior work. These results disprove many of the arguments the reviewer made.  For example, the reviewer suggested that hyperparameter sweeps are required when changing channel counts, but each dataset has a different number of channels: 38-Cloud dataset has 4, CHAMMI images are sourced from three datasets with 3, 4, and 5 channels that only partially overlap with each other, JUMP-CP has 8 channels, and So2Sat has 18. Yet each performs well with the same set of hyperparameters. These datasets also represent 3 different types of tasks (classification, segmentation, and representation learning). As such, they show broad generalization across a range of applications, contradicting the reviewer's assertion that they somehow do not.

---

### Official Review · Reviewer_La1T · 2025-07-02

**Clarity:** 3
**Significance:** 3
**Originality:** 2
**Rating:** 4
**Confidence:** 3

**Summary:**

This paper tackles a key limitation of applying Masked Autoencoders (MAEs) to Multi-Channel Imaging (MCI), where, unlike in standard RGB images, channels often contain complementary rather than redundant information. The authors argue that conventional patch-masking MAEs consequently fail to learn crucial cross-channel interactions in the MCI setting. They propose ChA-MAEVIT, a Vision Transformer framework that enhances cross-channel learning through a combination of four techniques: (1) a novel Dynamic Channel-Patch (DCP) Masking strategy that masks both random patches and entire channels; (2) Memory Tokens to serve as a global information store; (3) an efficient, single Channel-Aware Decoder that leverages channel-specific tokens for reconstruction; and (4) a Hybrid Token Fusion module for downstream tasks. Through extensive experiments on three diverse MCI datasets (CHAMMI, JUMP-CP, and So2Sat), the authors demonstrate that ChA-MAEVIT significantly outperforms state-of-the-art methods, particularly in scenarios with missing channels at test time.

**Questions:**

I'd appreciate the authors' feedback on the weakness section.

**Ethical Concerns:**

["NO or VERY MINOR ethics concerns only"]

**Final Justification:**

After reading the authors' response and the other reviewers' comments, I've decided to maintain my current rating.

**Limitations:**

Please see the weakness section.

**Quality:**

3

**Strengths And Weaknesses:**

Strengths:

1. The paper clearly articulates a fundamental mismatch between the assumptions of standard MAEs and the realities of MCI. The observation that patches in MCI primarily attend to their own channel under a standard patch-masking scheme (nicely visualized in Fig. 1) provides a compelling motivation for the work. Addressing this is significant as it can unlock the potential of powerful self-supervised learning for important scientific and industrial domains like microscopy and remote sensing.


2) The central idea of Dynamic Channel-Patch (DCP) Masking is both simple and highly effective. Forcing the model to reconstruct not just patches but entire channels using information from other available channels is a powerful pretext task that directly encourages the learning of cross-channel dependencies.


3) The experimental validation is thorough. The method is tested on three distinct and challenging MCI datasets with varying numbers of channels. The authors compare against a strong and relevant set of recent baselines, including the current SOTA (DiChaViT). The performance gains are substantial and consistently reported across datasets and settings.

Weaknesses:

1) While the DCP masking strategy is a clear and novel contribution, the other components are largely effective adaptations of existing ideas. Memory tokens are directly inspired by Register Tokens , the Channel-Aware Decoder builds upon the channel token concept from prior work like ChannelViT and DiChaViT, and the hybrid fusion module is a relatively standard approach.


2) Hyperparameter Complexity of DCP Masking: The proposed DCP masking introduces several new hyperparameters that appear sensitive. The appendix reveals that the two main configurations ("Combination" and "Alternate") require different settings for the patch ratio (0.25 vs 0.75, respectively). Furthermore, Figure 10 shows a non-trivial trade-off between performance on full vs. partial channel settings based on the balance between patch and channel masking. This suggests that applying ChA-MAEVIT to a new dataset might require careful and potentially expensive tuning of the masking strategy.

---

> ### Author Rebuttal · Authors · 2025-07-31
>
> We appreciate the reviewer’s thoughtful comments and respond to the questions below.
>
>
>
> > While the DCP masking strategy is a clear and novel contribution, the other components are largely effective adaptations of existing ideas. Memory tokens are directly inspired by Register Tokens, the Channel-Aware Decoder builds upon the channel token concept from prior work like ChannelViT and DiChaViT, and the hybrid fusion module is a relatively standard approach.
>
> Thank you for recognizing the novelty of our **DCP masking** strategy. We would like to clarify the contributions of the other components as follows.
>
> First, as stated throughout the paper, the **memory tokens** and **Hybrid Fusion module** are adapted from existing ideas but are employed for novel purposes. Specifically, we apply them in the context of multi-channel representation learning to enhance cross-channel interactions in a masked modeling framework. This constitutes a non-trivial shift in functionality: the memory tokens serve as global carriers of multi-channel semantics throughout the encoder-decoder pipeline. Hybrid Fusion module further contributes by making use of fine-grained patch tokens learned during reconstruction to enrich the final representation.
>
> Second, **Channel-Aware Decoder** introduces a new architectural design by integrating channel tokens with a lightweight shared decoder. This contrasts with prior work such as CA-MAE, which does not use channel tokens and instead relies on separate decoders for each channel. By enabling a unified decoding process, our approach improves scalability to a larger number of channels and yields better performance.
>
> Importantly, there is no guarantee that just combining prior ideas will work at all. For example, in Table 2 of the main paper, we show that integrating the strongest baseline (DiChaViT) with CA-MAE yields only modest improvements and still significantly falls short of ChA-MAEViT by 3.5%-12.5% across three datasets. Our work investigates the role of all the components, both existing, newly proposed, and their integration. The paper presents insights on what matters to solve open problems in the field.
>
> We will revise the camera-ready version to clarify these points.
>
> > Hyperparameter Complexity of DCP Masking: The proposed DCP masking introduces several new hyperparameters that appear sensitive. The appendix reveals that the two main configurations ("Combination" and "Alternate") require different settings for the patch ratio (0.25 vs 0.75, respectively).
>
> First, we would like to point out that the same masking ratios were used for every dataset in our evaluation, i.e., these hyperparameters are stable and generalize well to new settings, making them easy to use.  That said, these two settings use different hyperparameters, but result in the same amount of information being masked out in input images. In other words, the choice of patch ratio stems from ensuring the same amount of information is masked out, rather than the sensitivity of the hyperparameters.
>
> **Alternate** masking most closely resembles the prior work like MAE and CA-MAE and, thus,  we adopt a patch masking ratio of 0.75, consistent with standard practice for MAE and CA-MAE.
>
> **Combination** differs in that we mask entire channels, and, thus, we reduce the patch ratio to 0.25 due to the additional channel-level masking. As discussed in Appendix F.6 (line 652), a patch ratio of 0.25, when used with dynamic channel masking, results in approximately 65% of the spatial content being masked on average, which is comparable to the 75% masked in Alternate masking. Thus, applying a high patch ratio (e.g., 0.75) on top of channel masking would cause excessive information loss, making reconstruction extremely difficult and unstable. It is also reasonable to argue that masking entire channels is inherently more challenging than masking individual patches, as it removes all modality-specific information rather than local details, suggesting that a slightly smaller total masking ratio of 65% is warranted in this setting.
>
> > Furthermore, Figure 10 shows a non-trivial trade-off between performance on full vs. partial channel settings based on the balance between patch and channel masking. This suggests that applying ChA-MAEVIT to a new dataset might require careful and potentially expensive tuning of the masking strategy.
>
> As noted above ChA-MAEViT does **not** require expensive hyperparameter tuning as all experiments across all three datasets were performed using the same masking hyperparameters.  As shown in Table 4 of the main paper, both configurations significantly and consistently outperform the SOTA in Table 1 for both full and partial channel settings. Specifically, the Alternate configuration achieves boosts of 2.5-21.0%, while the Combination improves by 3.5-16.5%. This highlights the robustness and generalizability of the hyperparameters in our model across both satellite and microscopy imaging domains.
>
> Fig. 10 illustrates the effectiveness of combining both channels and patch masking. It shows that using both methods leads to better performance compared to using either one alone. Additionally, it provides insights into the trade-offs associated with each type of masking, which does not mean it requires costly tuning for new datasets.  Thus, we can recommend these masking ratios as reliable defaults for applying ChA-MAEViT to new datasets.

---

> > ### Comment · Reviewer_La1T · 2025-08-03
> >
> > I appreciate the authors' detailed response and would like to maintain my rating.

---

> > > ### Author Response · Authors · 2025-08-05
> > > **Official Comment by Authors**
> > >
> > > Dear Reviewer La1T,
> > >
> > > We appreciate your detailed review and positive feedback on our paper! We will ensure to incorporate the discussion into the final version.
> > >
> > > -Authors of paper 17675

---

### Official Review · Reviewer_7Rxw · 2025-07-02

**Clarity:** 3
**Significance:** 3
**Originality:** 2
**Rating:** 5
**Confidence:** 3

**Summary:**

The authors present four architecture improvements to the Masked Autoencoder (MAE) aimed at improving performance in multi channel with each channel capturing a semantically different aspect of the data. Such data such as it is frequently collected in scientific imaging, e.g. in fluorescence microscopy where different channels show different parts of the cell.

* Their new masking strategy does not only independently mask patches in different channels but additionally masks some channels completely. This is meant to encourage the model to capture inter-channel correlations as opposed to just intra-channel correlations.

* The authors introduce memory tokens, which are learned tokens that are meant to encode information about the data.

* The authors present an improved deocoder architecture.

* The authors present a new way to ustilize the learned representations in a downstream task by combining information from a CLS tokaen with a pooling operation arocss patch tokens.

**Questions:**

* The reconstructions in Figure 14, are missing high frequency details. Does this mean the information from the high frequency details is not encoded and thus is also not available for the classification task?

* Why is the "Dynamic Channel-Patch Masking" called "Dynamic"? This implies that the masking strategy somehow dynamically adapts to the data. I don't think this is the case. Am I missing something?

**Ethical Concerns:**

["NO or VERY MINOR ethics concerns only"]

**Final Justification:**

I think this is a very good paper. I am satisfied with the comments made by the authors and will stick to my positive rating.

**Limitations:**

Yes.

**Paper Formatting Concerns:**

None.

**Quality:**

3

**Strengths And Weaknesses:**

## Strengths ##

* I believe the development of novel architectures for multichannel data as it used in science does not get the attention it deserves. So this paper is addressing an urgent problem.

* I appreciate the wide range of different datasets the authors consider, showing the the method is generally applicable and not limited to a single application.

*  Even though the paper introduces a collection of new architectural features, the ablation studies investigate their efficacy separately, which is I think exactly the way it should be done to provide useful insights for the reader.

* The paper is well written.

* Results are substantially better than the baselines.

## Weaknesses ##

* This is really an engineering paper that presents improved architectures, with reasonable arguments for them but without a corresponding theory.

* Reconstruction quality in Figure 15 looks quite low. I understand that the goal of the paper is not the reconstruction but the learned representation; still I am not sure if this is to be expected or if related work produces higher quality results.

---

> ### Author Rebuttal · Authors · 2025-07-31
>
> Thank you for your insightful feedback! We address the reviewer’s questions below.
>
> > This is really an engineering paper that presents improved architectures, with reasonable arguments for them but without a corresponding theory
>
> Dynamic Channel-Patch (DCP) Masking involves removing both patches and entire channels from multi-channel inputs, where each channel typically contains distinct and complementary information, e.g., different sensor modalities or spectral bands. When these channels are combined, they often create emergent representations, which are features that arise solely from interactions between the channels. Masking disrupts not only the information within each channel but also the crucial synergy between channels.
>
> This concept can be formalized using mutual information. Let \$X_{\text{full}}$ represent the complete multi-channel input, and \$X_{\text{observed}}$ denote the unmasked subset. The relationship can be expressed as follows:
>
> $$
> I(Z; X_{\text{full}}) = I(Z; X_{\text{observed}}) + I(Z; X_{\text{masked}} \mid X_{\text{observed}})
> $$
>
> When part of the input is masked, the conditional mutual information term \$I(Z; X_{\text{masked}} \mid X_{\text{observed}})$ becomes zero, indicating that the model loses access to the synergistic cross-channel dependencies that could otherwise be recovered from the full input.
>
> Reconstruction loss, such as Mean Square Error, compels the model to predict the missing patches and channels:
>
> $$
> \mathcal{L}\_{\text{rec}} = \mathbb{E}\_{X} \left[ \sum_{c \in \mathcal{M}} \sum_{p} \left \| X\_{c,p} - \hat{X}_{c,p} \right \|_2^2 \right]
> $$
>
> Here, \$\mathcal{M}$ represents the set of masked channels, and \$p$ indexes spatial patches.
> This objective forces the encoder to capture both intra-channel structure and inter-channel dependencies, resulting in a more robust representation that retains emergent semantics and is resilient to missing modalities during inference.  We shall update these discussions in our camera ready.
>
>
>
> > Reconstruction quality in Figure 15 looks quite low. I understand that the goal of the paper is not the reconstruction but the learned representation; still I am not sure if this is to be expected or if related work produces higher quality results.
>
> Thank you for the insightful comment! Indeed, the low-fidelity reconstructions are expected and align with observations from prior work on Masked Autoencoders (MAEs), e.g., [A,B,C,D].
>
> As the reviewer noted, the primary objective of ChA-MAEViT is to learn high-quality representations, not to generate detailed reconstructions. Our design encourages the encoder to capture high-level semantic and cross-modal dependencies, while the decoder remains intentionally lightweight to verify whether essential latent information has been encoded. Additionally, following prior MAEs (e.g., [A,B,C,D]), we employ the mean squared error (MSE) loss for reconstruction, which often results in blurry outputs by averaging plausible solutions [D, E]. This leads to reconstructions dominated by low-frequency components, an expected outcome of the design choice.
>
> In contrast, models designed to generate high-fidelity images (e.g., [F, G, H]) typically rely on deeper, more expressive decoders. In such models, scaling the decoder is critical [H], and specialized loss functions such as perceptual loss [I] are employed to enhance visual quality.
>
> [A] Masked Autoencoders for Microscopy are Scalable Learners of Cellular Biology, Krau et al., CVPR’24
>
> [B] Masked Autoencoders Are Scalable Vision Learners, He et al., CVPR’22
>
> [C] Rethinking Transformers Pre-training for Multi-Spectral Satellite Imagery, Noman et al., CVPR’24
>
> [D] Scale-MAE: A Scale-Aware Masked Autoencoder for Multiscale Geospatial Representation Learning, Reed et aI., CCV’23
>
> [E] Training a Task-Specific Image Reconstruction Loss, Mustafa et al., WACV’22
>
> [F] Taming Transformers for High-Resolution Image Synthesis, Esser et al., CVPR’21
>
> [G] TokenFlow: Unified Image Tokenizer for Multimodal Understanding and Generation, Qu et al., CVPR’25
>
> [H] GigaTok: Scaling Visual Tokenizers to 3 Billion Parameters for Autoregressive Image Generation, Xiong et al., ICCV’25
>
> [I] The Unreasonable Effectiveness of Deep Features as a Perceptual Metric, Zhang et al., CVPR’18
>
> > Why is the "Dynamic Channel-Patch Masking" called "Dynamic"? This implies that the masking strategy somehow dynamically adapts to the data. I don't think this is the case. Am I missing something?
>
> The term "Dynamic" refers to the varying number of channels masked during training, rather than adaptation based on the specific input data. In other words, the masking strategy dynamically adjusts the number of masked channels and patches for each image, but this adjustment is independent of the actual content of the input samples. We will clarify this in the camera-ready version to avoid confusion.

---

> > ### Comment · Reviewer_7Rxw · 2025-08-04
> > **Post rebuttal discussion**
> >
> > I appreciate the reply by the authors and their clarifying explanations. I would like to stick with my initial rating.

---

### Official Review · Reviewer_4uNy · 2025-07-02

**Clarity:** 4
**Significance:** 3
**Originality:** 4
**Rating:** 5
**Confidence:** 5

**Summary:**

This paper introduces ChA-MAEViT, a new self-supervised training method for vision transformers, which addresses a flaw of random patch masking when applying masked autoencoders (MAE) to multi-channel imaging. Computer vision is typically developed for fixed channel configurations, but in multi-channel imaging, channels may provide unique complementary information for each image, and have low correlation with other channels. Random patch masking, does not explicitly optimise for learning information in one channel from another, which would be expected to provide robust representations for multi-channel images.

The authors highlight four key design choices for  ChA-MAEViT that address the challenge of applying MAEs to multi-channel imaging. The first is a new masking strategy. During training a subset of channels are masked out, in addition to masked patches from otherwise unmasked channels. All tokens from masked channels, and masked patches from unmasked channels, are reconstructed by the model via the model loss function. The second contribution are memory tokens. These are much like CLS or register tokens, learned embeddings appended to the input sequence, which are reported to act as long term memory. When channels are masked during training, these mask tokens may retain information about the missing channel from previous epochs/batches, that aid the reconstruction of a masked channel from the remaining others. Thirdly, ChA-MAEViT  uses a novel final representation, produced by aggregating the CLS token with all other patches, in what is referred to as hybrid token fusion. The last design choice that the authors highlight is a decoder which is shared across all channels. This is achieved by applying channel specific tokens to each patch embedding, to make the decoder aware of which channel each patch is derived from. This is crucial for training on datasets which contain large numbers of channels, as previous MAE approaches would consider a decoder-per channel which prevents scaling to datasets with many channels.

The paper compares ChA-MAEViT against a number of baselines across three multi-channel imaging datasets. For each dataset, it demonstrates superior performance in settings in which all channels are available at inference, and in which only some channels are available at inference.

A number of additional experiments are included to justify i) the inclusion of memory tokens, ii) their novel masking strategy, and iii) to demonstrate the utility of the shared channel-aware decoder.

**Questions:**

- Line 175 - can you please clarify what channel tokens were used in the encoder? The paper would read better if a brief description was included here.
- Could ChA-MAEViT be applied to consider OOD channels?
- Are there error bars that could be included in Table 2, Figure 3 and 4, to give an idea of statistical significance?

**Ethical Concerns:**

["NO or VERY MINOR ethics concerns only"]

**Final Justification:**

I initially felt that this paper should be accepted, due to the number of novel architectural choices introduced, the clarity with which they were explained, and the quality of the experiments to demonstrate their performance.

I had some minor comments about some results presented without statistical significance. The author's rebuttal has not caused any concern and has addressed my comments.

The authors have detailed responses to all other reviewers and have not changed my opinion of this paper. I therefore still recommend its acceptance.

**Limitations:**

Yes

**Paper Formatting Concerns:**

I have no major concerns.

**Quality:**

3

**Strengths And Weaknesses:**

This is a strong paper and is notable for its clarity, quality and significance. The paper is written to a high standard. The technical details are well described, and this ranks the paper high in terms of its reproducibility. There are many architectural choices introduced in this paper, and each of them are discernible due to the clear text and the care with which mathematical expressions are used and explained.

The choice of baselines are fair, in my opinion, and the superior performance of ChA-MAEViT is clear across each of the datasets, in the full and partial channel setting, as evidenced in Table 1. The quality of the experiments and the convincing performance gain with ChA-MAEViT provide the paper with significance. It remains an open challenge to develop robust representation learning for multi-channel imaging. This work has shown that random patch masking and MAEs are not purposely designed for the specific challenges associated with multi-channel data, where correlation between channels is often low. Recent works have scaled MAEs for multi-channel imaging in fluorescence microscopy, and this work points towards future work in which ChA-MAEViT is scaled, potentially leading to more robust representations for transfer learning. For these reasons, I think the community will be interested in the results presented here.

Where the paper is weaker is in the quality of some of the other experiments included. For example, while experiments are included which justify the use of channel masking, memory tokens and the shared decoder, these results, in Table 2 and Figs 3 and 4, lack measures of statistical significance. They would be improved with the addition of error bars. I also think the significance of the paper would be improved had the model been used to evaluate generalisation to OOD channels, as this is a persistent issue when training models with fluorescence microscopy data.


Overall, this paper is of high quality and I will recommend its acceptance.

---

> ### Author Rebuttal · Authors · 2025-07-31
>
> Thank you for your encouraging feedback! We address the reviewer’s questions below.
> > Are there error bars that could be included in Table 2, Figure 3 and 4, to give an idea of statistical significance?
>
> We report the mean and standard deviation (std) over three runs for the Table and Figures as follows for those results we were able to complete in the time constraints of the rebuttal period.
>
> **Table 2**
> | DiChaViT +        |  So2Sat (Full)       | So2Sat (Partial)      |
> |-------------------|------------------|--------------------|
> | SimCLR [10, 11]   |         64.44 ± 0.58     | 49.42 ± 0.63        |
> | SimSiam [12]                   | 64.07 ± 0.31      | 48.52 ± 0.71       |
> | iBOT [55]                       | 63.11 ± 0.32     | 47.84 ±  0.54       |
> | DINOv2 [38]                    | 63.42 ±  0.27    | 49.20 ± 0.72        |
> | MAE [23]                          | 62.88 ± 0.49     | 47.76 ± 0.62         |
> | **DCP Masking**  (ours)  | **66.02** ± 0.22   | **50.52** ± 0.45   |
>
> **Fig. 3b**
> | Number of Memory Tokens | CHAMMI Score |
> |-------------------------|--------------------|
> | 0                        | 73.42 ± 0.53     |
> | **4**                  | **74.63** ± 0.54 |
> | 8                       | 74.25 ± 0.40     |
> | 16                      | 74.53 ± 0.64     |
> | 24                      | 74.47 ± 0.68     |
>
> **Fig. 4**
> | Decoder Type                                                   | CHAMMI Score |
> |---------------------------------------------------------------|--------------------|
> | Shared Decoder w/out channel tokens             | 71.533 ± 0.47     |
> | Separate Decoders                                           | 72.339 ± 0.53     |
> | **Channel-Aware Decoder (Ours)**                  |  **74.63** ± 0.54    |
>
>
> The relatively low standard deviations (std) across experiments suggest the results are stable, supporting the robustness of our findings. We will include the corresponding error bars in the camera-ready version.
>
> >  Could ChA-MAEViT be applied to consider OOD channels, as this is a persistent issue when training models with fluorescence microscopy data.
>
> We completely agree that OOD channel generalization is a critical and ongoing challenge in fluorescence microscopy. In Appendix F.11 we demonstrate that ChA‑MAEViT can handle novel channels never seen during training in a preliminary experiment by adjusting its weights via learning to reconstruct the new channels at test time. Specifically, we adapt to novel channels by mapping them to existing ones using a few unlabeled examples and our self-supervised objectives, resulting in a 2.5% boost using just 640 examples.
>
> > Line 175 - can you please clarify what channel tokens were used in the encoder? The paper would read better if a brief description was included here.
>
> The channel tokens are learnable embeddings, each representing an input channel. Since all channels share a common projection layer, these tokens play a crucial role in capturing channel-specific features. They are concatenated with the spatial patch tokens and jointly processed by the transformer encoder and decoder. We will clarify this in the camera-ready version.

---

> > ### Author Response · Authors · 2025-08-05
> > **Official Comment by Authors**
> >
> > Dear Reviewer 4uNy,
> >
> > As we approach the end of the discussion period, we would greatly appreciate it if you could take a moment to read through our rebuttal. If you have any questions or need further clarification, we'd be happy to answer them. Thank you!
> >
> > -Authors of paper 17675

---

> > > ### Comment · Reviewer_4uNy · 2025-08-05
> > >
> > > Thanks to the authors for responding to each of my comments.
> > >
> > > Thanks for providing the additional results in the above response in the short rebuttal period. I agree with the authors, that this supports the robustness and significance of the results.
> > >
> > > Thank you also for pointing out the result in App F.11 that I had missed! The results, while outside the main scope of the paper, are very promising and a nice inclusion for the appendix.
> > >
> > > Thanks also for clarifying the channel embeddings used - the manuscript will be improved for its inclusion in the main text.
> > >
> > > Having read the other reviews, I am inclined to keep my score at accept.

---

### Decision · Program_Chairs · 2025-09-17

**Decision:**

Accept (poster)

**Comment:**

The paper proposes ChA-MAEViT, which augments MAEs for multi-channel imaging via dynamic channel-patch masking, memory tokens, hybrid token fusion, and a channel-aware shared decoder. Reviewers are largely positive (two Accepts, one BA), highlighting clear writing, strong results across CHAMMI, JUMP-CP, and So2Sat, and practical value when channels are missing; reported gains over prior MCI-ViTs are sizable. One reviewer remains concerned about conceptual incrementality and heuristic design, but the authors provided additional analyses that improve confidence: fixed default hyperparameters used across datasets (no per-dataset tuning) with consistent improvements, plus added experiments demonstrating segmentation performance and robustness under worst-case channel removal. Taken together, the method is sound, broadly applicable, and empirically strong; the AC recommends accept.